



# Synergistic radar and sub-millimeter radiometer retrievals of ice hydrometeors in mid-latitude frontal cloud systems

Simon Pfreundschuh[1], Stuart Fox[2], Patrick Eriksson[1], David Duncan[3], Stefan A. Buehler[4], Manfred Brath[4], Richard Cotton[2], and Florian Ewald[5]

[1]Department of Space, Earth and Environment, Chalmers University of Technology, 41296 Gothenburg, Sweden
[2]Met Office, FitzRoy Road, Exeter, EX1 3PB, United Kingdom
[3]ECMWF, Shinfield Park, Reading RG2 9AX, United Kingdom
[4]Meteorologisches Institut, Fachbereich Geowissenschaften, Centrum für Erdsystem und Nachhaltigkeitsforschung (CEN), Universität Hamburg, Bundesstraße 55, 20146 Hamburg, Germany
[5]Institut für Physik der Atmosphäre, Deutsches Zentrum für Luft- und Raumfahrt, Germany

**Correspondence:** Simon Pfreundschuh (simon.pfreundschuh@chalmers.se)

**Abstract.**

Accurate measurements of ice hydrometeors are required to improve the representation of clouds and precipitation in weather and climate models. In this study, a newly developed, synergistic retrieval algorithm that combines radar with passive millimeter and sub-millimeter observations is applied to observations of three frontally-generated, mid-latitude cloud systems in order to
validate the retrieval and asses its capabilities to constrain the properties of ice hydrometeors. To account for uncertainty in the assumed shapes of ice particles, the retrieval is run multiple times while the shape is varied. Good agreement with in situ measurements of ice water content and particle concentrations for particle maximum diameters larger than 200 µm is found for one of the flights for the Large Plate Aggregate and the 6-Bullet Rosette shapes. The variational retrieval fits the observations well although small systematic deviations are observed for some of the sub-millimeter channels pointing towards issues with
the sensor calibration or the modeling of gas absorption. We find that the quality of the fit to the observations is independent of the assumed ice particle shape, indicating that the employed combination of observations is insufficient to constrain the shape of ice particles in the observed clouds. Compared to a radar-only retrieval, the results show an improved sensitivity of the synergistic retrieval to the microphysical properties of ice hydrometeors at the base of the cloud.

Our findings indicate that the synergy between active and passive microwave observations improve remote-sensing measure-
ments of ice hydrometeors and may thus help to reduce uncertainties that affect currently available data products. Due to the increased sensitivity to their microphysical properties, the retrieval may also be a valuable tool to study ice hydrometeors in field campaigns. The good fits obtained to the observations increases confidence in the modeling of clouds in the Atmospheric Radiative Transfer Simulator and the corresponding single scattering database, which were used to implement the retrieval forward model. Our results demonstrate the suitability of these tools to produce realistic simulations for upcoming sub-millimeter
sensors such as the Ice Cloud Image or the Arctic Weather Satellite.





## 1 Introduction

The representation of clouds in climate models remains an important issue that causes significant uncertainties in their predictions (Zelinka et al., 2020). Improving and validating these models requires measurements that accurately characterize the distribution of hydrometeors in the atmosphere. At regional and global scales, such observations can be obtained efficiently

only through remote sensing. Unfortunately, currently available remote-sensing observations do not constrain the distribution of ice in the atmosphere well (Waliser et al., 2009; Eliasson et al., 2011; Duncan and Eriksson, 2018).

To address this, the Ice Cloud Imager (ICI) radiometer, to be launched onboard the second generation of European operational meteorological satellites (MetOp-SG), will be the first operational sensor to provide global observations of clouds at microwave frequencies exceeding 183 GHz. Compared to microwave observations at currently available frequencies ($\leq$

183 GHz), observations at and above 243 GHz have been shown to be sensitive to a broader size range of hydrometeors (Buehler et al., 2012) as well as their shape and particle size distribution (Evans et al., 1998). Although the increased sensitivity to smaller particles and their microphysical properties is expected to help improve measurements of ice in the atmosphere, it also increases the complexity of simulations of cloud observations, which are an essential tool for performing these measurements in the first place.

In Pfreundschuh et al. (2020), we have developed a cloud-ice retrieval based on radar and passive sub-millimeter observations to investigate the potential benefits of a synergistic radar mission to fly in constellation with ICI on MetOp-SG. The simulation-based results from Pfreundschuh et al. (2020) indicate that combining active and passive observations across millimeter and sub-millimeter observations can indeed help to better constrain the distributions of ice hydrometeors in cloud retrievals. The principal aim of this study is to validate the synergistic retrieval using real observations and to investigate its ability to retrieve

the vertical distributions of ice hydrometeors.

Since the retrieval has been shown to work on simulated observations, the validation of the synergistic retrieval essentially amounts to verifying the physical realism of the underlying forward model. The observations from the three flights considered here thus also constitute an opportunity to validate the radiative transfer model that is used in the retrieval, i.e. the Atmospheric Radiative Transfer Simulator (ARTS, Buehler et al., 2018) and the corresponding ARTS single scattering database (ARTS

SSDB, Eriksson et al., 2018), to accurately simulate cloud observations at sub-millimeter wavelengths. Such simulations are of paramount importance not only for future cloud retrievals from ICI observations (Eriksson et al., 2020) but also for assimilating cloud-contaminated observations in numerical weather prediction models (Geer et al., 2017).

In this study, the synergistic retrieval is applied to co-located radar and microwave radiometer observations of three mid-latitude cloud systems. The sensitivity of the retrieval to the ice particle shape that is assumed in the forward simulations is

tested by running the retrieval multiple times while varying the assumed shape. To test the accuracy of the retrieval, retrieval results are compared to in situ measurements of bulk ice water content (IWC) and particle size distributions (PSDs) for the two flights where these were available. Finally, we assess the consistency of the forward model simulations by investigating the agreement between simulated and real observations as well as between retrieved atmospheric state and in situ measurements.





The remainder of this article is structured as follows: Section 2 provides a description of the datasets and the retrieval

algorithm upon which this study is based. Section 3 presents the results of the retrieval as well as the comparisons to in situ data followed by a discussion of those results in Sec. 4 and conclusions in 5.

## 2   Data and methods

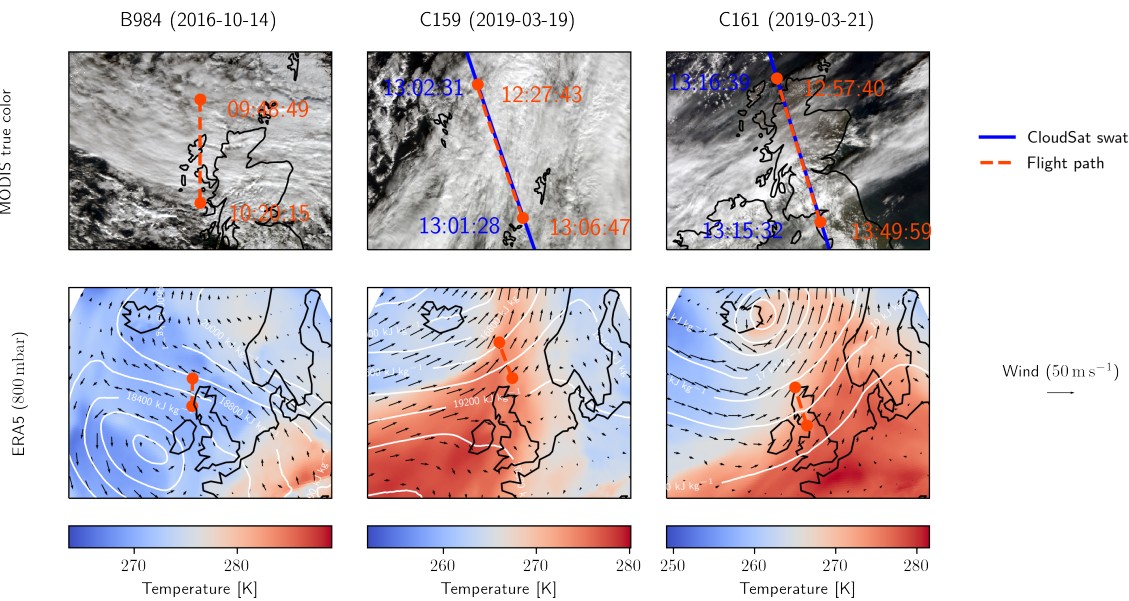

**Figure 1.** Flight paths of the cloud overpasses considered in this study. First row of panels shows the true-color composite derived from the closest overpasses of the MODIS (Team, 2017) sensor onboard the Aqua satellite. Second row shows ERA5 temperature (colored background), geopotential (contours) and wind speed (arrows) at the 800 mb pressure level from Hersbach et al. (2018).

The synergistic retrieval algorithm uses combined observations from radar and passive microwave sensors to retrieve vertical profiles ice hydrometeor distributions. The passive observations for this study are taken from the the MARSS (McGrath and

Hewison, 2001) and ISMAR (Fox et al., 2017) radiometers on board the UK's BAe-146-301 Atmospheric Research Aircraft (FAAM BAe-146) aircraft. Since the instrumentation of the FAAM BAe-146 aircraft does not include a cloud radar, only flights for which the radiometer observations can be co-located with radar observations from another platform are suitable for the combined retrieval. Since ISMAR is currently the only operational radiometer with channels up to 664 and 874 GHz, the flights considered in this study provide a rare opportunity to study the synergies between radar and passive (sub-)millimeter

observations using real observations.



**Table 1.** Datasets used in the this study.

| Title | Usage | Reference |
|---|---|---|
| HALO Microwave Package measurements during North Atlantic Waveguide and Downstream impact EXperiment (NAWDEX) | Radar observations for for flight B984 | Konow et al. (2018) |
| CloudSat 1B-CPR | Radar observations for flight C159 (Granule 67658), C161 (Granule 68702) | Tanelli et al. (2008) |
| FAAM B984 ISMAR and T-NAWDEX flight: Airborne atmospheric measurements from core instrument suite on board the BAE-146 aircraft | Radiometer observations and in situ measurements for flight B984 | Facility for Airborne Atmospheric Measurements (2016) |
| FAAM C159 PIKNMIX-F flight: Airborne atmospheric measurements from core and non-core instrument suites on board the BAE-146 aircraft | Radiometer observations and in situ measurements for flight C159 | Facility for Airborne Atmospheric Measurements (2019a) |
| FAAM C161 PIKNMIX-F flight: Airborne atmospheric measurements from core and non-core instrument suites on board the BAE-146 aircraft | Radiometer observations for flight C159 | Facility for Airborne Atmospheric Measurements (2019b) |
| ERA5 global reanalysis | A priori state and atmospheric background fields | Hersbach et al. (2018) |

An overview of the three flights and the corresponding meteorological contexts is provided in Fig. 1. The first considered flight, designated B984, took place on 14 October 2016 as part of the North Atlantic Waveguide and Downstream Impact Experiment (NAWDEX, Schäfler et al. (2018)). During this flight, a cloud system generated by an occluded front has been observed quasi-simultaneously by three research aircraft: The High Altitude and Long Range Research Aircraft (HALO, Krautstrunk and Giez (2012)), the FAAM BAe-146 and the Falcon 20 of the Service des Avions Francais Instrumentations pour la Recherche en Environnement (SAFIRE). The two other flights, designated C159 and C161, took place in March 2019 as part of the PIKNMIX-F campaign. These two flights were performed following the ground track of simultaneous overpasses of the CloudSat satellite. The observations probe clouds in different regions of a frontal system generated by a low-pressure system passing over Iceland around 21 March 2019. The cloud system observed during flight C159 is a stratiform, lightly-precipitating cloud located in the warm sector of the frontal system, whereas the clouds observed during flight C161 are of convective origin and located in the active region of the cold front. All datasets that were used in this study are listed together with their sources in Tab. 1.





## 2.1 Radar observations

The radar observations from the three flights are displayed in Fig. 2. Observations for flight B984 stem from the HAMP MIRA
radar (Mech et al., 2014) onboard the HALO aircraft, which operates at a frequency of 35 GHz and has been characterized and
calibrated by Ewald et al. (2019). Its observations have been downsampled to a vertical resolution of 210 m and a horizontal
resolution of roughly 700 m in order to reduce the computational complexity of the retrieval and to better match the field of
view of the passive observations, which, at an altitude of 5 km, vary between about 900 m for the low- frequency channels and
200 m for the high-frequency channels.

The radar observations for flights C159 and C161 stem from the CloudSat Cloud Profiling Radar (CPR, Tanelli et al. (2008)),
which operates at 94 GHz. Since the CloudSat observations were affected strongly by ground-clutter, the first five bins located
completely above surface altitude were set to the reflectivity found in the sixth bin above the surface. The CPR observations
have a vertical resolution of 240 m and a horizontal resolution of about 1.4 km. The horizontal distance between subsequent
observations is 1.1 km.

### 2.1.1 MARSS

The MARSS radiometer measures microwave radiances at 89 GHz, 157 GHz and channels located around the water vapor line
at 183 GHz. Although MARSS is a scanning radiometer only observations within 5 ° off nadir are used in the retrieval. The
observations from the three flights are displayed in Fig. 3. Observations from channels that are sensitive to surface emission
(89 GHz and 157 GHz) are excluded from the retrieval for flight sections over land. The MARSS observations were mapped
to the radar observations using nearest-neighbor interpolation.

### 2.1.2 ISMAR

The ISMAR radiometer has channels covering the frequency range from 118 GHz up to 874 GHz. As for MARSS, only
observations within 5° degrees off nadir are used in the retrieval. The observations from the 3 flights are displayed in Fig. 4.
Similar as for the two low-frequency channels of MARSS, the 4 outermost channels around the 118 GHz oxygen line are not
used over land. The matching of ISMAR observations to radar observations is performed in the same way as for MARSS. It
should be noted that not all channels were available on all flights: The channels around 448 GHz were not available on the
B984 flight, while the two of the channels around 325 GHz were missing for the C159 and C161 flights. From the channels at
874 GHz only the V polarization was available for flights C159 and C161.

The polarized measurements at 243 GHz and at 664 GHz for flight B984 were replaced by the average of the measured H
and V polarizations. For flights C159 and C161, only the horizontally-polarized measurements at 664 GHz were used used due
to excessive noise in the V channel.



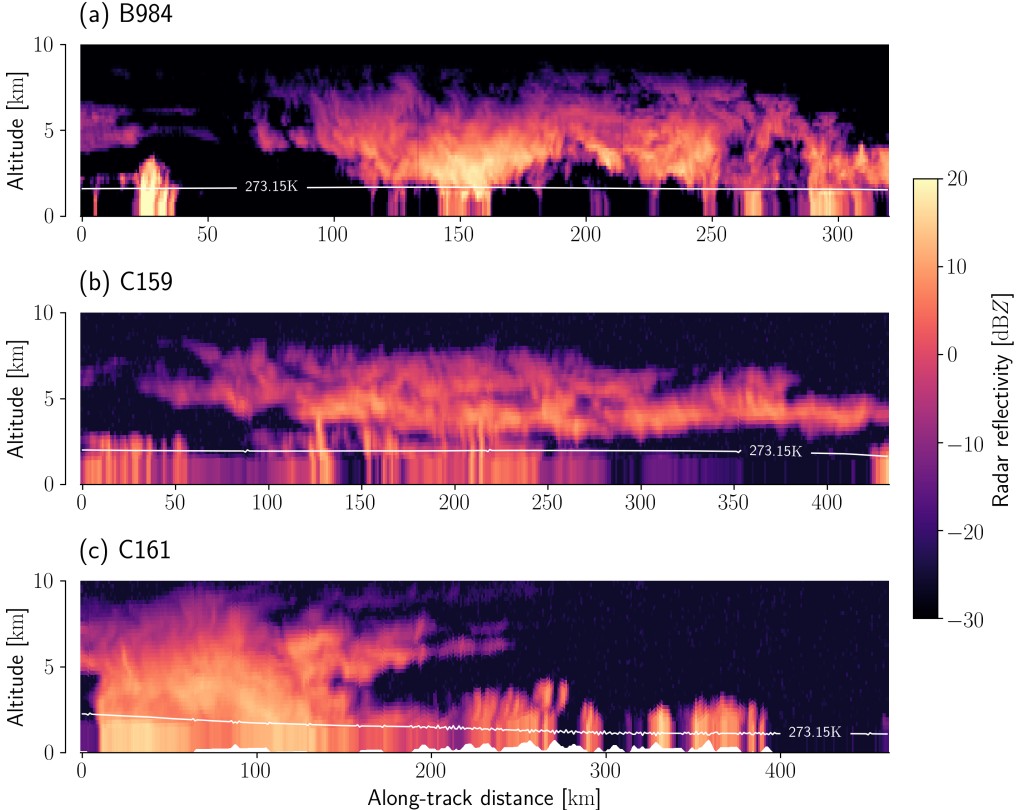

**Figure 2.** Radar observations from the flights used in this study. Panel (a) shows the radar reflectivity measured by the HAMP MIRA 35 GHz cloud radar. Panels (b) and (c) show the reflectivity measured by the CloudSat CPR at 94 GHz. The white line displays the ERA5 freezing level from Hersbach et al. (2018).

## 2.2   in situ measurements

in situ measurements of cloud hydrometeors were performed during the flights B984 and C159. A detailed view of the high-level runs and the corresponding in situ sampling paths are provided in Fig. 5. For flight C159, this view reveals a noticeable horizontal offset of 3 to 4 km between the ground track of radar and radiometer observations and even larger deviations between the lower parts of the in situ sampling flight path and the high-level run.

The in situ measurements that are relevant to this study are bulk ice water content measured using a Nevzorov hot-wire probe (Korolev et al., 2013) and PSDs recorded using DMT CIP-15 and CIP-100 probes, which measure size-resolved particle concentrations with resolutions of 15 and 100 μm, respectively. The in situ measurements were mapped to corresponding radar observations using a nearest-neighbor criterion.





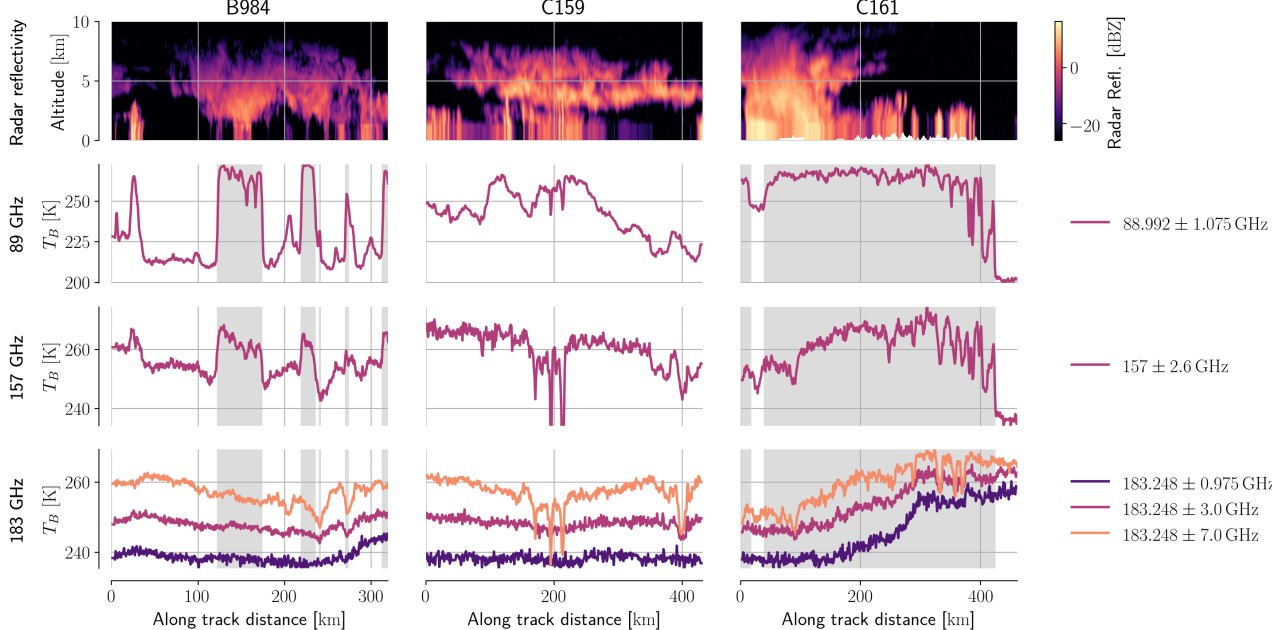

**Figure 3.** Passive microwave measurements from the MARSS radiometer together with the matched radar observations. Grey background in the radiance plots marks observations that were taken over land.

An overview of the measured IWC and PSDs is provided in Fig. 6. While for flight B984 the measured IWC are mostly consistent with the radar observations, there are clear disparities between the measured IWC and the CPR reflectivities for flight C159. This indicates that there may be considerable differences between the regions of the cloud that were sampled during the in situ sampling and the part the was observed by the CloudSat CPR.

The PSD profiles for flight B984 show a clear size-sorting pattern with a gradual decrease of the concentration of particles smaller than 200 μm and a simultaneous increase of the concentration of larger particles. For flight C159, high concentrations of small particles are encountered at low altitudes which decrease with altitude. For larger particles no systematic variation with altitude is observed.

## 2.3 Retrieval algorithm

The synergistic retrieval algorithm used in this study is based on the optimal estimation framework (Rodgers, 2000) and retrieves distributions of frozen and liquid hydrometeors together with water vapor by simultaneously fitting a forward model to the active and passive observations. Since the algorithm is described in detail in Pfreundschuh et al. (2020) the following section only outlines its main features and how it has been adapted to the flight data.

The retrieval input consists of a single radar profile and the corresponding spatially closest radiometer observations. Background properties of atmosphere and surface, such as temperature and wind speed, as well as a priori profiles for relative





**Figure 4.** Passive microwave measurements from the ISMAR radiometer together with the matched radar observations. Grey background in the radiance plots marks observations that were taken over land and are therefore not used in the retrieval.

humidity and liquid cloud water are taken from the ERA5 hourly reanalysis (Hersbach et al., 2018). The output of the retrieval are two parameters of the PSDs of frozen and liquid hydrometeors as well as liquid cloud water content (LCWC) and relative humidity. All retrieval targets and corresponding a priori assumptions are listed in Tab. 2.

The retrieval forward model has been adapted to the sensors that were available for each flight. The atmospheric grid was
limited to altitudes between 0 and 10 km and matched to the resolution of the radar observations. The latest stable release (version 2.4) of the Atmospheric Radiative Transfer Simulator (ARTS, Buehler et al. (2018)) is used to implement the forward model used in the retrieval. The built-in single-scattering radar solver of ARTS is used to calculate radar observations and





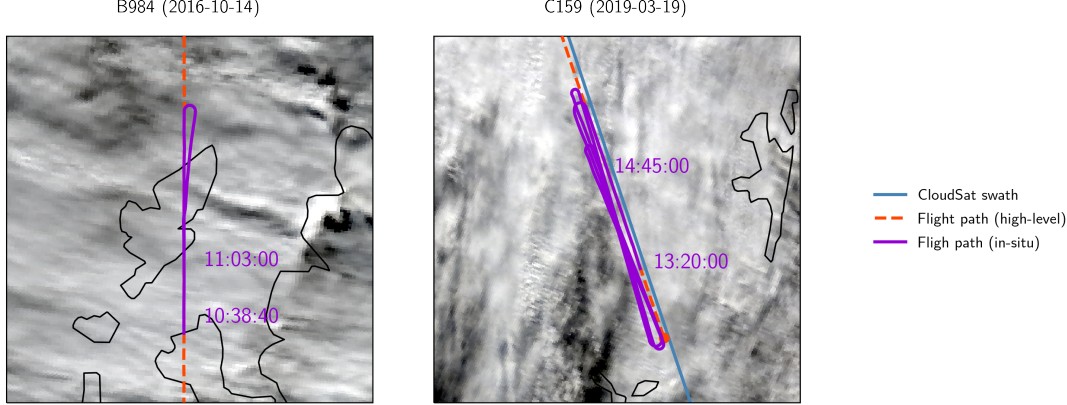

**Figure 5.** Detailed view of the flight paths of the high-level runs and in situ sampling paths for flights B984 and C159. The backound is the true-color composite derived from the closest overpasses of the MODIS (Team, 2017) sensor on the Aqua satellite.

**Table 2.** Retrieval quantities and a priori assumptions used in the retrieval. The relation for the a priori mean of $\log_{10}(N_0^*)$ is taken from (Cazenave et al., 2019).

| Quantity | Retrieved parameters | A priori mean | A priori std. dev. |
|---|---|---|---|
| Ice water content (IWC) | $\log_{10}(N_0^*)$ | $-0.076586\cdot(T-273.15)+17.948$ with $T$ temperature in K | 2 |
| | $D_m$ | $\text{IWC} = 10^{-6}$ | 500 µm |
| Rain water content (RWC) | $\log_{10}(N_0^*)$ | 7 | 2 |
| | $D_m$ | 500 µm | 500 µm |
| Cloud liquid water content (CLWC) | $\log_{10}(\text{CLWC})$ | From ERA5 | 1 |
| Relative humidity (RH) | $\text{arctanh}(\frac{2\cdot\text{RH}}{1.1} - 1.0)$ | From ERA5 | 1 |

Jacobians. To account for the effect of multiple scattering in CloudSat observations, the attenuation due to hydrometeors is scaled at each atmospheric layer by a factor of 0.5 following Fig. 16 in Battaglia et al. (2010). Passive radiances are calculated using the ARTS interface to DISORT (Stamnes et al., 2000) and their Jacobians are approximated using a first order scattering approximation. Gaseous absorption is modeled using the absorption models from Rosenkranz (1993) for $N_2$ and $O_2$. Following



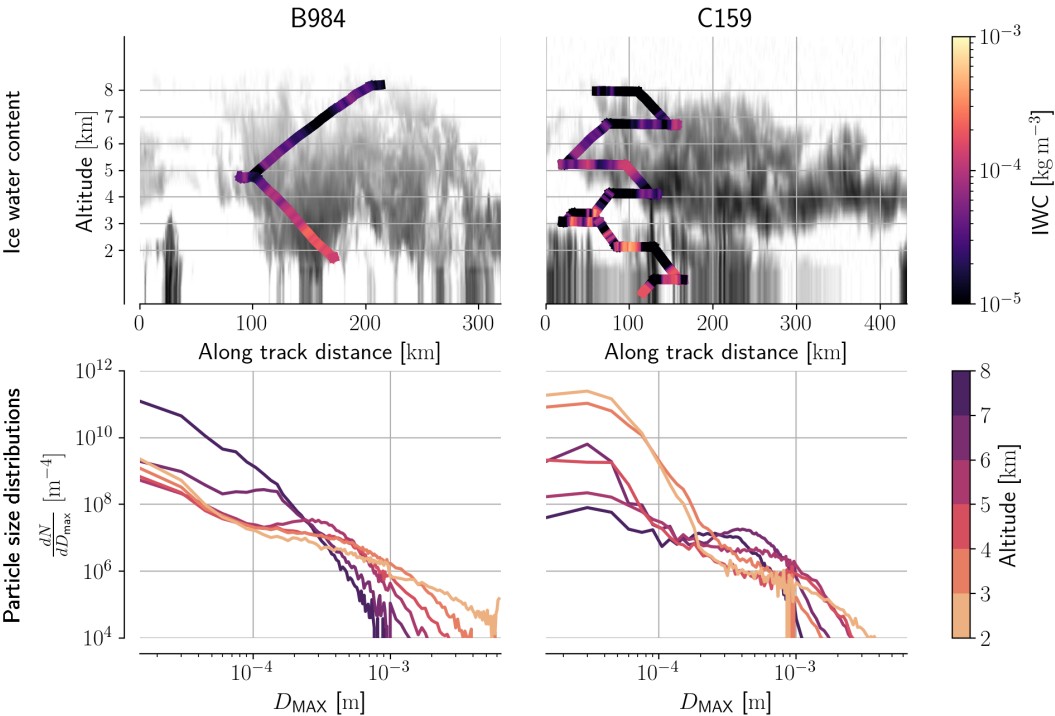

**Figure 6.** in situ measured IWC and PSDs for flights B984 and C159. The first row of panels displays the measured IWC along the flight path plotted on top of the co-located radar observations. The second row displays the variation of the mean of the in situ measured PSDs for different altitudes in the cloud.

Fox (2020), absorption from water varpor is calculated using a combination of the AER database v3.6 (Cady-Pereira et al., 2020) for resonant absorption and the MT-CKD model version 3.2 for continuum absorption (Mlawer et al., 2012).

### 2.3.1 Representation of frozen hydrometeors

The forward model simulates active and passive observations in two steps: In the first one, the bulk properties that are used to represent hydrometeors in the retrieval are mapped to corresponding optical properties. The optical properties are then, in the second step, used together with background atmosphere and surface to simulate the observations.

The mapping of bulk to optical properties is based on a PSD and an ice particle habit that associates particles of different sizes and shapes to optical properties. More specifically, the forward model uses the normalized PSD approach proposed by

Delanoë et al. (2005) with the mass-weighted mean diameter ($D_m$) and intercept parameter ($N_0^*$) as parameters. The updated values from Cazenave et al. (2019) are used as shape parameters of the distribution. The ice particle habit is represented by a collection of ice particle shapes together with corresponding, pre-computed optical properties. Bulk optical properties are calculated by integrating the product of particle density and optical properties over the particle size. As the retrieval is currently





set up, the particle habit cannot be retrieved and must be assumed a priori. Since this is difficult, a set of habits has been chosen
with which the retrieval will be run in order to investigate the impact of the selected habit on the results.

Five particles were selected from the set of standard habits that is distributed with the ARTS SSDB (Eriksson et al., 2018).
The standard habits are particle mixes that combine pristine crystals at small sizes with aggregate shapes at larger sizes. The
selected habits are listed in Tab. 3. To provide an overview of their optical properties, characteristic bulk optical properties have
been calculated and displayed in Fig. 7 together with their mass-size relationships. The PSD used to calculate the bulk optical
properties is the same that is used in the retrieval with the $N_0^*$ value set to the a priori value at a temperature of 260 K. The
particles were selected so that their properties cover most of the variability of the available set of standard habits both in terms
of the mass-size relationship as well as their optical properties.

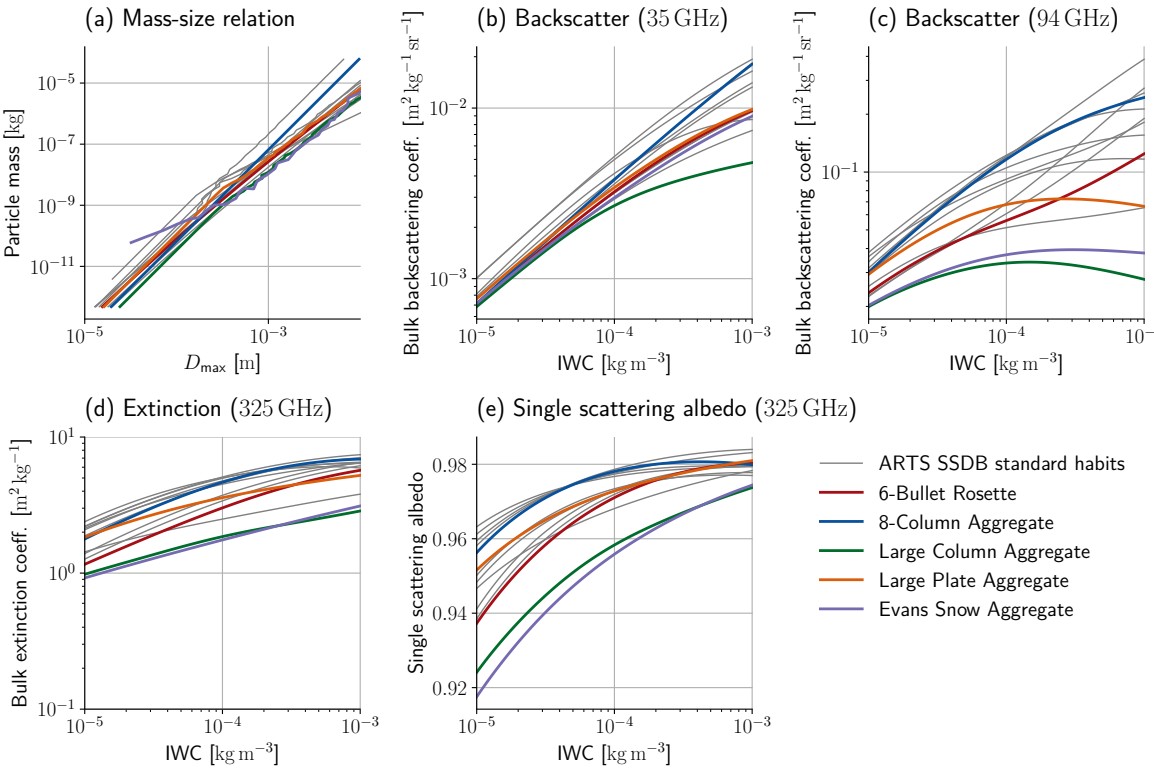

**Figure 7.** Properties of the selected ice particle shapes that are used to represent frozen hydrometeors in the retrieval forward model. Colored
lines display the properties of the selected habits, while grey lines show the properties of the remaining standard habits distributed with the
ARTS SSDB. Bulk optical properties were calculated using the PSD parametrization that is used in the retrieval.

Complementary information that can help guide the selection of a suitable particle shape can be obtained from in situ
measurements. Since the particle habit associates particle sizes with a specific shape it can be used to compute a bulk water
content corresponding to PSD measurements. This allows calculating the IWC corresponding to the in situ measured PSDs,





**Table 3.** Particle habits used in the retrieval. The mass size relationship is given in terms of the parameters of a fitted power law of the form $m = \alpha \cdot D_{\mathrm{MAX}}^{\beta}$ with $D_{\mathrm{MAX}}$ the maximum diameter and $m$ in kg.

| Habit name | Shapes used | Size range | | Mass size relationship | |
|---|---|---|---|---|---|
| | Name (ID) | $D_{\mathrm{eq,\,min}}$ [µm] | $D_{\mathrm{eq,\,max}}$ [µm] | $\alpha$ | $\beta$ |
| 6-Bullet Rosette | 6-Bullet Rosette (6) | 16 | 8905 | 0.4927 | 2.4278 |
| 8-Column Aggregate | 8-Column Aggregate (8) | 10 | 3000 | 440 | 3 |
| Evans Snow Aggregate | Evans Snow Aggregate (1) | 50 | 2109 | 0.196 | 2.386 |
| Large Plate Aggregate | Thick Plate (15) | 16 | 200 | 110 | 3 |
| | Large Plate Aggregate (20) | 160 | 3021 | 0.21 | 2.26 |
| Large Column Aggregate | Block Column (12) | 10 | 200 | 110 | 3 |
| | Large Column Aggregate (18) | 160 | 3021 | 0.25 | 2.43 |

which can be compared with the IWC measured by the Nevzorov probe. The agreement between the PSD-derived IWC and the in situ measured IWC can then provide insight into whether the mass-size relation corresponding to the particle shape is consistent with that of the particles in the cloud. Such a comparison is provided in Fig. 8.

For both flights, the Large Plate Aggregate and the 6-Bullet Rosette yield the best overall agreement with the in situ measured
IWC. The Large Column Aggregate yields values at the low end of the measured distribution for all flights and altitudes. The Evans Snow Aggregate yields similar results to those of the Large Column Aggregate except at high altitudes for flight B984 and low altitudes for flight C159. The 8-Column Aggregate generally yields higher IWC values than most other habits and tends to overestimate the in situ IWC at low altitudes for flight B984.

## 3 Results

The primary results of the combined retrieval are the retrieved hydrometeor distributions. In addition to that, the retrieval also fits a radiative transfer model to the observations whose agreement with the real observations can provide valuable information regarding the accuracy of the forward model and the fitness of a priori and modeling assumptions.

### 3.1 Fit to observations

The retrieval residuals, i.e. the difference between simulated and real observations, for the Large Plate Aggregate habit are
displayed in Fig. 9. As the figure shows, the retrieval was able to fit both radar and radiometer observations fairly well for all flights. For flight B984, the radar residuals show some scattered deviations located at the edge of the cloud, which are likely discretization artifacts. Except for that, residuals for this flight remain well within 1 dB. The residuals for flight C159, exhibit four vertical stripes with significant residuals in the radar observations. In these regions, which correspond to significant scattering depressions in most passive channels up to 325 GHz, the simulations overestimate the radar reflectivity. Apart from
this, there are some smaller regions where the simulations underestimate the radar reflectivity but these remain limited to



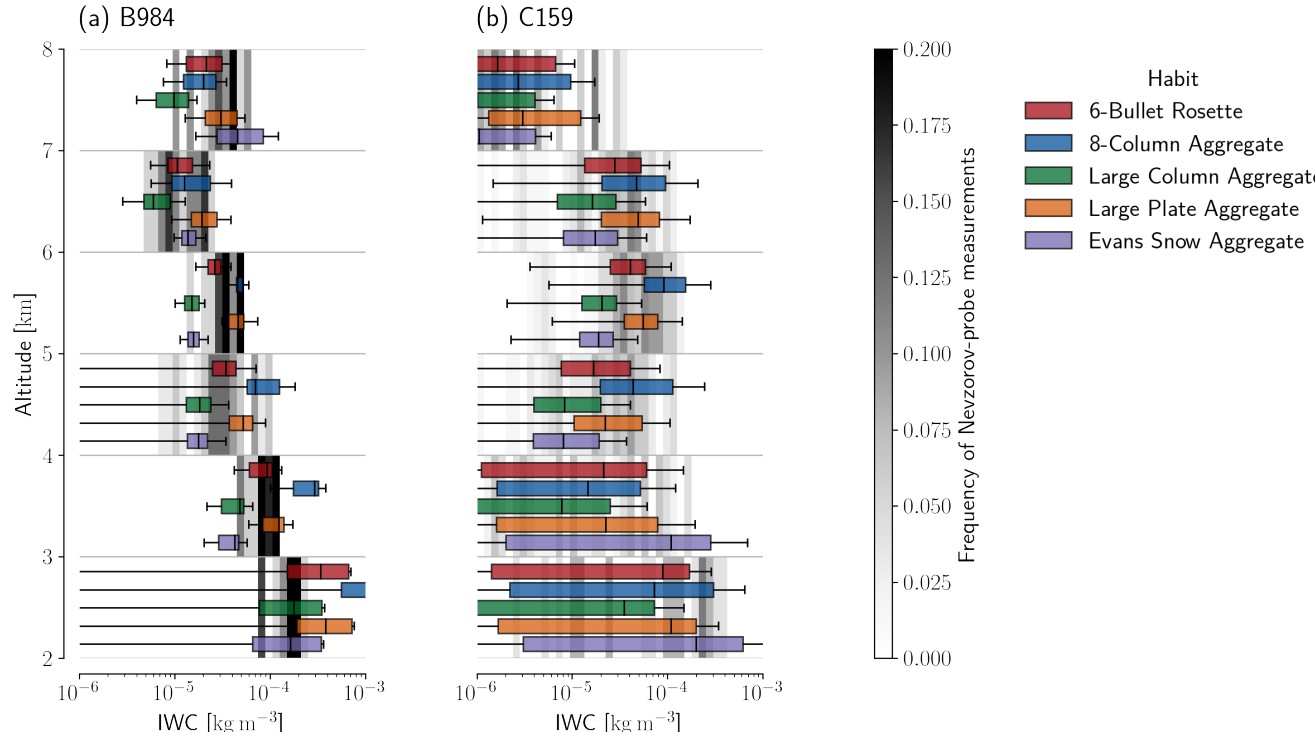

**Figure 8.** Comparison of bulk IWC as measured by the Nevzorov probe and inferred from in situ measured PSDs using a given particle shape. The background in each plot shows the distribution of Nevzorov-measured IWC for a given 1 km-altitude bin. Colored boxes display the distribution of IWC in that bin inferred for a given particle shape. Boxes, whiskers and outliers are drawn following Tukey's conventions for box plots.

within few dB. For flight C161, moderate negative residuals in the radar observations can be observed in the right half of the convective core, which coincide with an overestimation of the scattering signal at 243 GHz.

Radiometer residuals for flight B984 are mostly within ±5 K. For the two other flights the residuals are larger. Differences up to and exceeding 10 K are observed at the window channels up to 243 GHz as well as in the outermost channels around the
absorption lines at 118 GHz and 183 GHz. Since these correspond to profiles in which residuals of opposite sign are present in the radar observations, a likely explanation is that they are caused by small-scale precipitation events that are missed by one of the sensors due to co-location issues.

For a more systematic analysis of the effect of the assumed particle shape on the retrieval residuals, their distribution for radar and radiometer channels around 183 GHz and above are displayed in Fig. 10. The distributions, which for most channels
are close to or centered around zero, confirm that the retrieval generally fits the observations well. The largest deviations are observed for the 874 GHz channel and the 243 GHz channel for flight C161. For flights C159 and C161, the 874 GHz and 664 GHz channels exhibit small systematic biases of opposite signs, which may indicate issues with the calibration or the





**Figure 9.** Differences between observed and fitted observations for the Large Plate Aggregate particle. First two rows depict the radar observations and their residuals, respectively. Following rows show the retrieval residual in the radiometer measurements for each of the frequency bands used in the retrieval. The grey shading marks sections of the flight path that were located over land surfaces.





modeling of water vapor absorption at these channels. Furthermore, it is interesting to note that the ice particle habit only has a minor impact on the residuals indicating that the retrieval can compensate for mismatches in the assumed particle shape by

adjusting the retrieval variables.

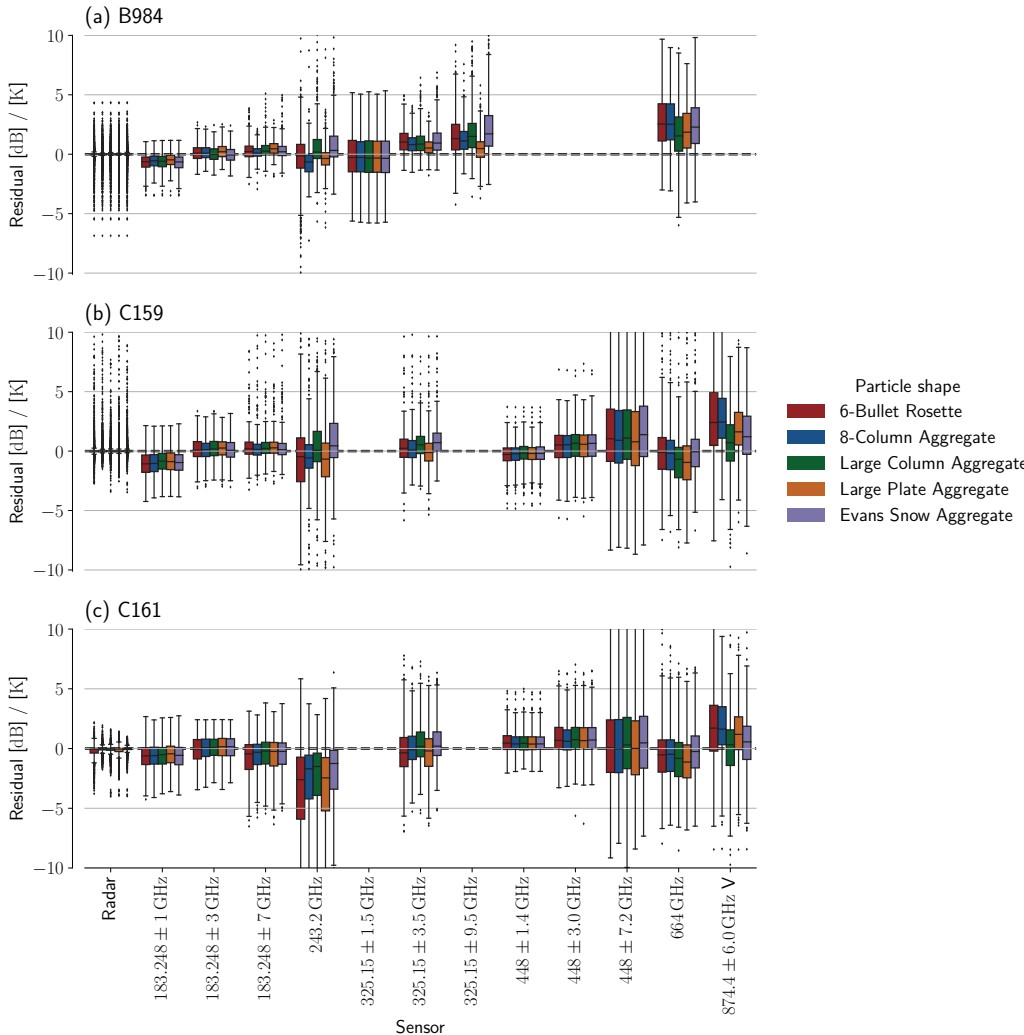

**Figure 10.** Distributions of retrieval residuals for different particle shapes used in the forward model for each of the three flights.

## 3.2   Retrieved ice water content

The retrieved bulk IWC and corresponding IWP for all three cloud scenes are displayed in Fig. 11. For all three flights, the ice particle shape has a significant effect on the retrieved amount of ice. In terms of IWP, the Large-Column Aggregate and Evans Snow Aggregate habits yield the highest values, while the 8-Column Aggregate consistently yields the lowest IWP. The Large





Plate Aggregate and 6-Bullet Rosette both yield values within the range of the other particle models with the 6-Bullet Rosette leading to slightly higher IWP values than the Large Plate Aggregate. In addition to the effect of the increased total retrieved water content, the particle habit also has a small effect on the vertical distribution of the ice hydrometeors, which is visible particularly for retrieved IWC in the convective core observed during flight C161.

**Figure 11.** Retrieved IWP and IWC content for all flights and different ice particle shapes assumed in the retrieval. The white line displays the ERA5 freezing level from Hersbach et al. (2018).





## 3.3 Comparison to in situ measurements

The most important question regarding the hydrometeor retrieval is certainly whether the retrieved bulk properties are consistent with the in situ measurements. As mentioned in Sec. 2.2, the in situ measurements were mapped to the radar observations using a nearest-neighbor criterion. For B984, retrieval results within a distance of 1km of the flight path were then associated to the in situ measurements. Because of the mismatch between observations and in situ sampling paths, another approach was taken for flight C159. Here the retrieval results were mapped to the in situ measurements by selecting all retrieval results between $50$

and $150$ km along-track distance. Both, the matched retrieval results and the vertically-resolved distributions of measured and retrieved IWC are displayed in Fig. 12.

    For flight B984, the distribution of in situ measured IWC values is well within the range of retrieved IWC values across all particle shapes up to an altitude of around $7$ km. At these altitudes, the best match to the in situ measurements is achieved with the 6-Bullet Rosette particle and the Large Plate Aggregate. The 8-column aggregate underestimates the in situ-measured

IWC while Large Column Aggregate and Evans Snow Aggregate overestimate it. Above $7$ km all particles lead to results that underestimate the in situ measured IWC. A likely cause for this is the high concentration of small particles as observed in the in situ measurements (c.f. Fig. 6) for which microwave observations lack sensitivity.

    For flight C159, the distribution of retrieved IWC still covers the distribution of in situ measured values for altitudes above $3$ km but exhibits a tendency towards underestimation. Overall, the differences between the results for different habits are

smaller for this flight. However, the uncertainties caused by the large sampling region as well as the potential co-location issues affecting the results make them less conclusive.

    Finally, we want to address the question whether the representation of cloud microphysics within the retrieval forward model is consistent with the in situ measured PSDs. For this, we calculate the PSDs corresponding to the retrieved bulk properties and compare them to the in situ measurements. The results of the comparison are displayed in Fig. 13. For flight B984, we

find good agreement between retrieved and in situ measured PSDs for larger particles ($D_{\mathrm{MAX}} > 200$ μm) for the Large Plate Aggregate and the 6-Bullet Rosette. Since this is observed even at altitudes above $6$ km, it confirms that the underestimation of IWC at these altitudes is likely caused by the high concentration of smaller ice particles. For flight C159, the retrieved PSDs deviate significantly from the in situ measurements. Although the 6-Bullet Rosette and Large Plate Aggregate seem to fit the tail ($D_{\mathrm{MAX}} > 1$mm) of the PSD for most altitudes except between $3-4$ km, the measured PSDs deviate considerably at smaller

sizes. This may also indicate that the the assumed shape of the PSD may not be suitable for the observed cloud system.

## 4   Discussion

This study used a novel, synergistic retrieval to retrieve vertically-resolved distributions of ice hydrometeors from co-located radar and microwave radiometer observations. For most of the considered channels, the retrieval succeeded in fitting both the active and passive observations without significant, systematic deviations. For two of the flights, the retrieved hydrometeor

distributions were compared to in situ measurements. For one of the flights (B984), we found good agreement with the in situ measured bulk IWC for altitudes between 2 and 6 km for two of the particle shapes. The same particle shapes also yield



**Figure 12.** in situ measured and retrieved IWC for flights B984 and C159. The first row of panels shows the in situ sampling paths in relation to the measured radar reflectivity as well as the retrieval values that are mapped to the in situ measurements (blue shading). The second row of panels shows in the background the distribution of in situ measured IWC values in altitude bins with a height of 500 m. Colored boxes display the corresponding distribution of retrieved IWC values for different ice particle shapes. Boxes are drawn following Tukey's conventions for box plots. The colored triangles mark the mean of the distribution.



**Figure 13.** in situ measured and retrieved PSDs for flights B984 (left column) and C159 (right column). Each row of panels shows the mean of the in situ measured PSDs (black) together with randomly drawn samples of measured PSDs (light grey) for a given altitude bin of a height of one kilometer. Colored lines on top show the corresponding mean retrieved PSD for different assumed particle shapes.



the best agreements with the in situ measured PSDs for larger ice particles ($D_{\mathrm{MAX}} > 200\ \mu\mathrm{m}$). For the second flight (C159), no consistency was found between the in situ-measurements and the retrieval. A likely explanation for this is that the in situ measurements are not as well co-located due to the temporal and spatial differences between the different observations as well as the in situ measurements.

### 4.1 Sub-millimeter radiative transfer in cloudy atmospheres

A first important result of this study is the ability of the retrieval to find atmospheric states that are consistent with the observed radiances and radar reflectivities for all three flights. This in itself is not self-evident due to the uncertainties that still affect the modeling of ice-particle scattering at millimeter and sub-millimeter wavelengths. Previous studies that tried to directly validate sub-millimeter RT through clouds were either limited to tropical clouds (Evans et al., 2005; Eriksson et al., 2007) or cirrus clouds (Fox et al., 2017). For flight B984, the radar and all passive observations were fitted up to small systematic biases no larger than 3 K. The deviations for the two other flights were generally larger, but these were likely caused by spatial and temporal co-location errors. This indicates that both the assumed optical properties as well as the retrieval forward model are consistent across the considered wavelengths. Furthermore, the two particle shapes for which he best agreement between retrieved and in situ measured hydrometeor distributions was found for flight B984, were also those whose mass-size relationship yielded the best agreement between in situ measurements of IWC and PSDs. Since this ties the microphysical properties of the particles to their optical properties, it suggests that the modeling of these particles in the ARTS SSDB is physically consistent.

### 4.2 The impact of assumed ice particle shape

A rather unexpected result that emerged from this study is that the retrieval can fit the observations regardless of the assumed ice particle shape. This indicates that although the observations are sensitive to variations in ice particle shape, they alone cannot constrain it. This is in agreement with what has been reported in Pfreundschuh et al. (2020), namely that no correlation could be found between the particle shape yielding best retrieval fit and the one yielding the most accurate retrieval results.

In an effort to better separate a potential signal from the ice particle shape in the retrieval residuals, Fig. A1 in the appendix displays the relation between IWP and corresponding residuals in the $325 \pm 3.5\,\mathrm{GHz}$ channel. This channel was chosen because it belongs to the channels displaying the largest differences between residual distributions for different particle habits (Fig. 10). Nonetheless, the plots exhibit no sign of a relationship between either residual and ice water content or the residuals across different habits.

This result implies that future ice hydrometeor retrievals that make use of millimeter and sub-millimeter microwave observations must either account for the uncertainty caused by variations in ice particle shape or find ways to more accurately constrain the shape a priori. Moreover, for studies that seek to validate model predictions by comparing simulated and observed microwave observations, this implies that care must be taken to accurately characterize the ice particle shape. This is because consistency between simulations and observations can be achieved for bulk water contents that vary by almost an order of magnitude (c.f. Fig. 11).





### 4.3 Representation of cloud microphysics

The lack of a signal that constrains the ice particle shape even in the combined observations puts additional weight on the question of how to best represent ice particles in simulations of microwave observations. The habits that lead to the most accurate retrieval results in this study were the Large Plate Aggregate and the 6-Bullet Rosette. It is interesting to note that the Large Plate Aggregate was also found to yield the best agreement between NWP-model-based simulations and satellite observations at frequencies between 19 and 190 GHz for stratiform snow in Geer (2021).

Nonetheless, these findings are based on observations from the single flight for which the retrieval results could be reliably compared with in situ measurements. This result can thus be seen as indication that these habits may work well for similar mid-latitude cloud systems but more generally applicable conclusions would require further and more systematic investigation.

### 4.4 Retrieval validation

Since the results presented in Pfreundschuh et al. (2020) were limited to simulations based on a high-resolution climate model, the validation of the retrieval using real observations remained an open issue. For flight, B984 good agreement was found between retrieval results and in situ measurements. Although the retrieved IWC deviates from the in situ measurements at altitudes $> 6$ km, the retrieved PSDs still match the in situ measurements well for particles with $D_{\mathrm{MAX}} > 200$ μm. This indicates that the observed deviations are due to the presence of a large number of small ice particles that the microwave observations are not sensitive to. Although O'Shea et al. (2021) and O'Shea et al. (2019) show that the occurence of high particle concentrations at sizes below 200 μm may be due to measurement inaccuracies of the CIP-15 probe, the measured PSDs correctly reproduce the measured IWC at these altitudes when the corresponding water content is calculated using any of the tested particle habits (Fig. 8). Furthermore, the presence of a cloud layer with a large number of small particles was also reported by Ewald et al. (2021) who investigated the same cloud system with combined radar-lidar observations.

For flight C159 no good agreement was found between retrieved and in situ measured IWC and PSDs. However, some evidence suggest that this may be due to co-location: Firstly, the flight path for the in situ sampling was found to be offset from the high-level run during which the observations were taken in the direction opposite to the wind at 800 mb (Fig. 1, Fig. 5). Secondly, the a clear backscattering signal is present in the CloudSat CPR observations even in regions where only negligible amounts of IWC are present in the in situ measurements (Fig. 12). Thirdly, the residuals observed in Fig. 9 are indicative of additional co-location issues between radar and radiometer observations. Finally, also the comparison of retrieved and in situ measured PSDs (Fig. 13) seems to indicate large deviations between the observed and the assumed PSD shape.

### 4.5 The added value synergistic cloud retrievals

Although the evidence from flight B984 suggests that the synergistic retrieval algorithm works well for retrieving ice hydrometeor concentrations, similar retrievals can be performed using only radar observations. A retrieval using only radar observations has the obvious advantage of requiring only a single sensor and being computationally much less complex. This naturally leads to the question of the added value that a synergistic retrieval can provide.





To investigate this, the results of the combined and an equivalent radar-only retrieval for flight B984 are displayed in Fig. 14. For the Large Plate Aggregate and 6-Bullet-Rosette habits, the results of the combined and the radar-only retrieval are largely similar down to an altitude of about $3.5$ km below which the radar-only retrieval tends to overestimate the in situ IWC. In

310 contrast to the combined retrieval, the results of the radar-only retrieval exhibit almost no impact from the particle habit. So while the radar-only retrieval remains mostly unaffected by the habit choice, using a different habit in the combined retrieval may cause systematic overestimation (Evans Snow Aggregate and Large Column Aggregate) or underestimation (8-Column Aggregate).

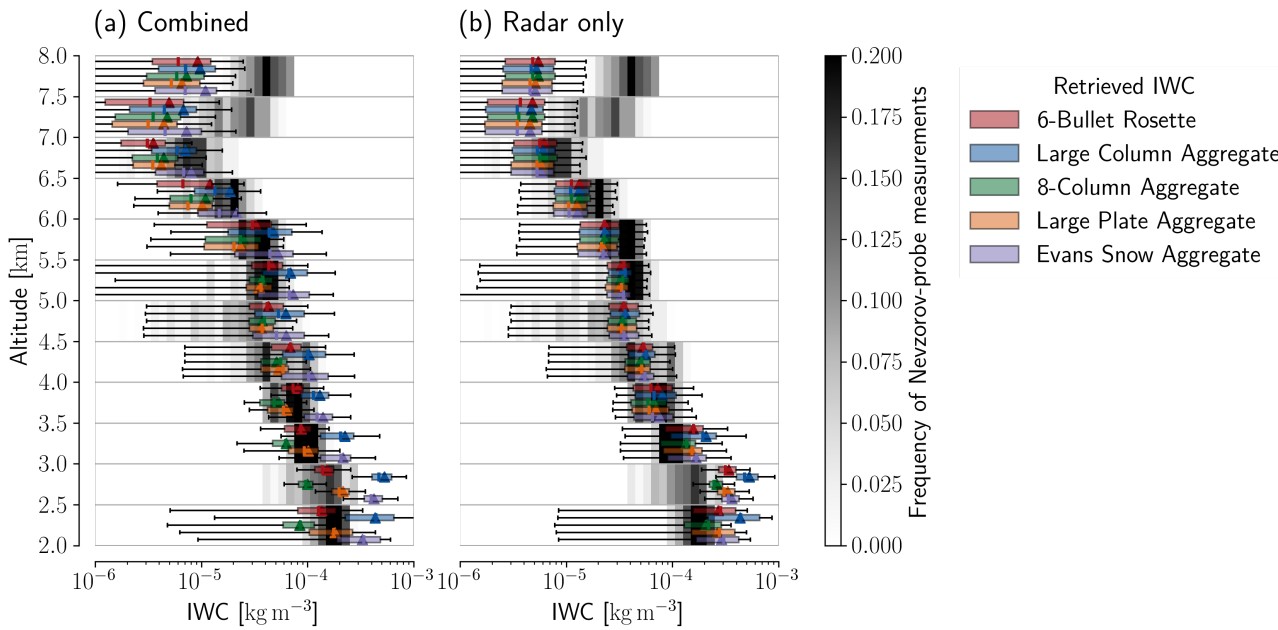

**Figure 14.** in situ measured and retrieved PSDs for flight B984 for the combined and the radar-only retrieval. Each panel displays the distributions of in situ measured IWC for different altitude bins in the background and the distribution of retrieved IWC values for different ice habits as colored boxes on top.

A similar comparison is shown in Fig. 15 for the retrieved PSDs. The PSDs are largely similar for both retrievals for alti-

315 tudes above $4$ km. Below that, however, the radar-only retrieval overestimates the particle concentrations, while the combined retrieval matches the in situ measurements well for the Large Plate Aggregate and 6-Bullet Rosette habits. This indicates that the combined retrieval utilizes the complementary information in the radar and passive observations to match both moments of the PSD, whereas the radar-only retrieval can only match one of them.





**Figure 15.** in situ measured and retrieved PSDs for flight B984 retrieved using the combined (panel (a)) and the radar-only retrieval (panel (b)). Each row of panels shows the mean of the in situ measured PSDs (black) together with randomly drawn samples of measured PSDs (light grey) for a given altitude bin of a height of one kilometer. Colored lines on top show the corresponding mean retrieved PSD for different assumed particle shapes.





These results thus suggest that combining radar with passive microwave observations helps to constrain the PSD of ice hy-
drometeors for sufficiently large particle sizes ($D_{\mathrm{MAX}} > 200\,\mu$m). Since for air- and space-borne observations only microwave
observations can sense the base of thick clouds, this is a unique synergy between these types of observations.

### 4.6 Limitations

Since microwave radiative transfer simulations in cloudy atmospheres remain a challenging problem, it is important to also
consider the limitations of the simulations and derived results that were presented in this study. The simplifications that were
applied in the simulations are the following:

1. Horizontal photon transport between the retrieved profiles is ignored.

2. Inhomogeneity across the radar and radiometer beams is ignored.

3. The finite spectral resolution of the passive channels is neglected.

4. The radar solver neglects multiple scattering.

5. The effects of particle orientation are ignored.

Barlakas and Eriksson (2020) found that neglecting photon transport in simulations of sub-millimeter observations across a
footprint of $6\,\mathrm{km}$ incurs only a small random error with biases $< 0.5\,\mathrm{K}$, so it is likely also small for the simulations presented
here. For flight B984, the horizontal averaging of the radar observations leads to a profile width of $700\,\mathrm{m}$ which is fairly close
to the width of the radiometer field of views which varies between about $900$ and $200\,\mathrm{m}$ at an altitude of $5\,\mathrm{km}$. The effect
of beam inhomogeneity is therefore expected to be small for this flight. For flights C159 and C161, the radar beam has an
along-track width of about $1.4\,\mathrm{km}$, which is larger than that of the radiometers, so these observations may be affected to a
larger extent than those for flight B984.

Neglecting the finite spectral resolution of the passive channels can lead to an error of up to $2.1\,\mathrm{K}$ for satellite observations
that are affected by Ozone absorption (Eriksson et al., 2020). Since the passive observations used in this study were all taken
from altitudes below $10\,\mathrm{km}$ the effect of this approximation is likely negligible.

The effect of multiple scattering for air-borne radar observations is generally negligible (Battaglia et al., 2010). For CloudSat
observations, however, the higher frequency and the considerably wider footprint will increase the effects of multiple scattering
on the observations. Although the simulations account for the signal-enhancing effect of multiple scattering by layer-wise
reduction of the attenuation, the presence of multiple scattering may still add to the uncertainty in the simulations for flights
C159 and C161.

Finally, there is the potential presence of oriented particles in the cloud. The different ice habits used in this study all assume
totally random orientation of the ice particles. Systematic vertical orientation of particles of a given shape in the cloud would
effectively alter their scattering properties. For observations at nadir, particle orientation can increase the extinction of the
Large Plate Aggregate of up to $20\,\%$ (Barlakas et al., 2021). To first order, the increase in extinction can be expected to cause





a similar overestimation of the retrieved IWC. This, however, is still considerably lower than the differences observed due to different ice habits.

## 5 Conclusions

The main result from the experiments presented in this study is that we were able to find two ice particle shapes, the Large Plate Aggregate and the 6-Bullet Rosette, for which the results of the combined retrieval were consistent with the observations

as well as the in situ measured IWC and PSDs for flight B984. Considering the co-location issues that likely affected the other two flights, we interpret this as a cautious indication of the validity of the retrieval implementation. Since the ARTS radiative transfer model and optical properties from the ARTS single-scattering database constitute a crucial component of the retrieval, this result also indicates that they work reliably across the millimeter- and sub-millimeter domain.

The results confirm the simulation-based findings from (Pfreundschuh et al., 2020), that a synergistic retrieval based on active

and passive microwave observations can help to better characterize the PSD of large ice hydrometeors ($D_{\mathrm{MAX}} > 200\,\mu m$) than a radar-only retrieval alone. This indicates that such retrievals can be used to study the microphysical properties of clouds and thus help to improve their representation in weather and climate models.

However, the retrieval is at the same time very sensitive to the assumed ice particle habit that is used in the retrieval forward model. We found no evidence of a signal that could help to constrain the ice particle shape based solely on the combination

of radar and microwave observations, not even when sub-millimeter observations are included. This means that more work is needed to better constrain the shape a priori or that even more observations must be integrated into the retrieval.

Although further work will be required, this study demonstrates the feasibility and potential of synergistic retrievals of ice hydrometeors by combining active and passive observations at millimeter and sub-millimeter wavelengths. Since the combined retrieval can better constrain the PSD of ice hydrometeors, it may be a useful tool to study the representation of clouds in

NWP and climate models. Additionally, as illustrated in this study, the retrieval can be used to study the representation of ice hydrometeors in radiative transfer simulations, which will be vital to many applications of observations from upcoming sub-millimeter sensors such as ICI and the Arctic Weather Satellite (ESA, 2021).

*Code availability.* All code used to produce the results in this study is available through public repositories (Simon Pfreundschuh, 2019; Pfreundschuh, 2021).

*Data availability.* A detailed listing of the datasets that were used in this study together with their sources is provided in Tab. 1.





**Figure A1.** Scatter plots of retrieved IWP and corresponding residual in the fitted observations $325 \pm 3.5\,\mathrm{GHz}$ ISMAR channel. Each column displays the residual distributions for the five different particle habits.



*Author contributions.* Simon Pfreundschuh has performed the retrieval calculations and data analysis as well as written the manuscript. Patrick Eriksson, Stefan A. Buehler, Manfred Brath, David Duncan and Simon Pfreundschuh have collaborated on the study that lead to the development of the presented algorithm. Stuart Fox, Richard Cotton, Florian Ewald have provided the flight campaign data, guidance regarding their usage and contributed to the interpretation and discussion of the retrieval results.

*Competing interests.* No competing interests are present

*Acknowledgements.* The work of SP and PE on this study was financially supported by the Swedish National Space Agency (SNSA) under grants 150/14 and 166/18.

SB was supported by the Deutsche Forschungsgemeinschaft (DFG, German Research Foundation) under Germany's Excellence Strategy — EXC 2037 'Climate, Climatic Change, and Society' — Project Number: 390683824, contributing to the Center for Earth System Research
and Sustainability (CEN) of Universität Hamburg.

SB's work contributes to the Cluster of Excellence "CLICCS—Climate, Climatic Change, and Society" funded by the Deutsche Forschungsgemeinschaft DFG (EXC 2037, Project Number 390683824), and to the Center for Earth System Research and Sustainability (CEN) of Universität Hamburg."

The computations for this study were performed using several freely available programming languages and software packages, most
prominently the Python language (The Python Language Foundation, 2018), the IPython computing environment (Perez and Granger, 2007), the numpy (van der Walt et al., 2011), pandas (Reback et al., 2021) and xarray (Hoyer and Hamman, 2017) packages for numerical computing, the satpy package (Raspaud et al., 2021) for processing of satellite data and matplotlib for generating figures (Hunter, 2007).

The computations were performed on resources at Chalmers Centre for Computational Science and Engineering (C3SE) provided by the Swedish National Infrastructure for Computing (SNIC).
Hersbach et al. (2018) was downloaded from the Copernicus Climate Change Service (C3S) Climate Data Store.

The results contain modified Copernicus Climate Change Service information 2021. Neither the European Commission nor ECMWF is responsible for any use that may be made of the Copernicus information or data it contains.

HALO data are from SPP 1294 "High Altitude and Long Range Research Aircraft (HALO)", funded by the German Research Foundation (Deutsche Forschungsgemeinschaft - DFG), project number 316646266.



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
