# Peer review of "Synergistic radar and sub-millimeter radiometer retrievals of ice hydrometeors in mid-latitude frontal cloud systems"

_Atmospheric Measurement Techniques, 2021_

## Author Comment (AC2)

**1 Comments from reviewer 2**

We would like to thank the reviewer for reading our manuscript and providing helpful feedback.

**1.1 Specific comments**

**Reviewer comment 1**

Lines 69-71. It may be helpful to add the time periods of these campaigns in the text or in the table.

**Author response:**

We will include the requested information in the revised version of the manuscript.

**Changes in manuscript**

- The sentence introducing the B984 flight will be modified to:

  **Changes starting in line 66:**

  The first considered flight, designated B984,  was performed on 14 October 2016  during the North Atlantic Waveguide and Downstream Impact Experiment (NAWDEX), which took place during September and October 2016 (Schäfler et al., 2018).

- The sentence introducing the C159 and C161 flights will be modified to:

  **Changes starting in line 71:**

  The two other flights, designated C159 and C161,  were part of the PIKNMIX-F campaign, which took place in March 2019.

**Reviewer comment 2**

Line 100. It seems that you do not have the same measurements to perform the retrievals for different flights. How is the missing information handled in the retrieval method?

**Author response:**

To handle the varying availability of channels (and sensors) across the different flights, the retrieval implementation is adaptive in the sense that it can be run with arbitrary sensor configurations. Channels that are sensitive to surface emission and thus only used over Ocean are disabled by setting their assumed uncertainties to very high values.

**Changes in manuscript:**

- To make this clearer, we will reformulate the sentence that describes the adaption of the forward model and retrieval.

  > **Changes starting in line 134:**
  >
  > The  forward model and retrieval were made adaptive so that the ingested observations can be easily adapted to the  different sensors and channels that were available for each flight. Low frequency channels that are used only over Ocean surfaces are deactivated over land by setting the corresponding channel uncertainty to $10^6$ K.

**Reviewer comment 3**

Figure 10. The discrepancies are quite large for 243 GHz channel for flight C161 compared to the other two flights. Could you comment on that?

**Author response:**

As we explain in the manuscript, we suspect that the remaining discrepancies in the 243 GHz channel for flights C159 and C161 are mainly caused by precipitation that may be observed differently by the different sensors due to co-location issues. This reasoning is based on the observation that similar residuals are observed in the same location in other passive channels that are sensitive to the lower atmosphere as well as the radar. For the region where the largest biases are observed for flight C161 the temporal delay between radiometer and radar observations is about 30 minutes during which the structure of the cloud has likely changed thus leading to inconsistencies between the radar and radiometer observations.

The initial version of the manuscript has not mentioned the temporal co-location of the observations that differs significantly between the flights. For the revised manuscript we propose to add a new figure that displays the time delay between the radar and radiometer observations for the different flights. We will also extend the discussion of the residuals.

**Changes in manuscript:**

- We will add the figure shown in Fig. 1.1 together with the description shown below to the the section the presents the radar observations.

**Changes starting in line 90:**

While the radar observations for flight B984 come from an airborne radar, the observations for flights C159 and C161 stem from a spaceborne sensor. The high velocity of the spaceborne sensor causes significant temporal delay between co-located observations from the radiometers and the radar. Figure 1.1 displays the delay between co-located radar and radiometer observations with respect to the along-track distance for the three flight scenes. So while the delays for flight B984 remain mostly within 5 minutes, they reach values exceeding 30 minutes for the two other flights.

- The presentation of the retrieval residuals will be extend as follows.

**Changes starting in line 188:**

Radiometer residuals for flight B984 are mostly within ±5 K  but larger for flights C159 and C161. For these two flights, residuals exceeding 10 K are observed  in the window channels up to 243 GHz as well as  the outermost channels around the absorption lines at 118 GHz and 183 GHz. Since these  occur in profiles where precipitation is present and in which similar residuals can be observed in the radar observations, a likely explanation is that they are caused by  precipitation that is not observed by all sensors due to spatial and temporal co-location issues. Especially the large residuals in the 243 GHz channel for flight C161 at around 100 km along track distance may well be caused by the evolution of the convective cloud during the delay of almost 30 minutes that separates the radiometer and radar observations.

**Reviewer comment 4**

Lines 220-222. As you also mentioned, the largest uncertainties correspond to the higher-level clouds where we have smaller ice particles. Are these uncertainties also related to the lack of representativeness of the particle shape/type used in the models?

**Author response:**

As we meant to express in the paragraph starting in l. 285, our hypothesis is that the underestimation of IWC high up in the cloud that is observed across all tested particle models is rather due to the mismatch in shape between the assumed and observed PSD. Since none of the sensors used in this study has any significant sensitivity to particles with diameters smaller than 200 $\mu$m, the contribution of those particles to the total

[Figure]

Figure 1.1: Delays between the co-located observations from radar and radiometers for the three flights.

IWC is essentially inferred through the assumed shape of the PSD. Since the assumed PSD drastically underestimates the amount of those particles, this may explain why the retrieved IWC is lower than the true IWC.

**Changes in manuscript:**

To make this point clearer, we will reformulate the paragraph that discusses the deviations from the in situ measurements.

> **Changes starting in line 288:**
>
> This indicates that the  concentrations of these larger particles may be retrieved correctly but that the total IWC is underestimated due to the mismatch between assumed and actual PSD shape, the former of which lacks the very high concentration of small particles that are present in the in situ measurements. Although O'Shea et al. (2021) and O'Shea et al. (2019) show that the  occurrence of high particle concentrations at sizes below 200 $\mu$m may be due to measurement inaccuracies of the CIP-15 probe, the measured PSDs correctly reproduce the measured IWC at these altitudes when the corresponding water content is calculated using any of the tested particle habits (Fig. 8).

**Bibliography**

O'Shea, S., Crosier, J., Dorsey, J., Gallagher, L., Schledewitz, W., Bower, K., Schlenczek, O., Borrmann, S., Cotton, R., Westbrook, C., et al.: Characterising optical array particle imaging probes: implications for small-ice-crystal observations, Atmospheric Measurement Techniques, 14, 1917–1939, 2021.

O'Shea, S. J., Crosier, J., Dorsey, J., Schledewitz, W., Crawford, I., Borrmann, S., Cotton, R., and Bansemer, A.: Revisiting particle sizing using greyscale optical array probes: evaluation using laboratory experiments and synthetic data, Atmospheric Measurement Techniques, 12, 3067–3079, 2019.

Schäfler, A., Craig, G., Wernli, H., Arbogast, P., Doyle, J. D., McTaggart-Cowan, R., Methven, J., Rivière, G., Ament, F., Boettcher, M., Bramberger, M., Cazenave, Q., Cotton, R., Crewell, S., Delanoë, J., Dörnbrack, A., Ehrlich, A., Ewald, F., Fix, A., Grams, C. M., Gray, S. L., Grob, H., Groß, S., Hagen, M., Harvey, B., Hirsch, L., Jacob, M., Kölling, T., Konow, H., Lemmerz, C., Lux, O., Magnusson, L., Mayer, B., Mech, M., Moore, R., Pelon, J., Quinting, J., Rahm, S., Rapp, M., Rautenhaus, M., Reitebuch, O., Reynolds, C. A., Sodemann, H., Spengler, T., Vaughan, G., Wendisch, M., Wirth, M., Witschas, B., Wolf, K., and Zinner, T.: The North Atlantic Waveguide and Downstream Impact Experiment, Bull. Amer. Met. Soc., 99, 1607–1637, https://doi.org/10.1175/BAMS-D-17-0003.1, 2018.

---

## Author Response (AR1)

**Response to comments**

Synergistic radar and sub-millimeter radiometer retrievals of ice hydrometeors in mid-latitude frontal cloud systems

This document contains the responses to the comments of each reviewer followed by the marked-up differences of the manuscript and the revised version. For each comment the author's response and, if applicable, the corresponding changes in the manuscript are listed. Line numbers of changes are given with respect to the revised manuscript.

The two most prominent changes in the revised manuscript are:

- Addition of a figure showing the temporal delay between the active and passive observations, which we suppose explains the larger residuals for the C159 and C161 flight. Extension of the discussion of the residuals for those flights to mention the temporal offsets between the active and passive observations.

- Addition of a figure showing scatter plots of IWP and residuals of the $243\pm2.5$ GHz and $325 \pm 9.5$ GHz channels for flight B984, which hint at a weak relationship particle shape and residuals for this flight.

**1 Comments from reviewer 1**

We would like to thank the reviewer for reading our manuscript and providing helpful feedback.

**1.1 Specific comments**

**Reviewer comment 1**

Line 87: Although it probably doesn't matter much to the passive frequencies being simulated, it would make more sense to me to extend the retrieved hydrometeor content at the sixth bin above the surface downward rather than the reflectivity, since the reflectivity is likely not constant (due to attenuation).

**Author response:**

We would like to thank the reviewer for this thoughtful comment that draws attention to an aspect of the retrievals that we may have paid too little attention to. It is of course true, that the assumption of constant reflectivity through the ground clutter region down to the surface is a very crude one. The liquid hydrometeors retrieved in this region affect the retrieval of frozen hydrometeors aloft through the radiative background of the passive observations and forcing the retrieval to fit the constant reflectivity down to the surface may thus affect the retrieved concentrations of ice hydrometeors.

To investigate the effect of this simplification, we suggest a slightly different approach than the one proposed by the reviewer: Instead of enforcing constant hydrometeor concentrations from the upper boundary of the ground clutter region down to the surface, we retain the constant reflectivities from before but set the uncertainties associated with the affected bins to infinity. This instructs the retrieval to ignore the reflectivities from those bins but retains the possibility of adapting the concentrations of liquid hydrometeors to be consistent with the passive observations.

Fig. 1.1 and Fig. 1.2 show the effect of this change in the retrieval setup on the observation residuals as well as the retrieved ice water content. The only prominent change in the results is the misfit in the radar bins that were excluded from the retrieval. Retrieval residuals and retrieved concentrations of ice hydrometeors remained unaffected. We thus conclude that this design decision, albeit questionable, has only an insignificant effect on the results presented here.

**Reviewer comment 2**

Table 2: For the Dm corresponding to IWC, what is the meaning of the "A priori mean" value of "IWC $= 10^{-6}$"? Shouldn't the a priori mean be in units of length (as with RWC)?

**Author response:**

The a priori mean for $D_m$ was chosen so that the corresponding ice water content has a constant value of $10^{-6}$ kg m$^{-3}$ throughout the atmosphere.

**Changes in manuscript:**

To make this clearer, we will change the entry in the table from '$IWC = 10^{-6}$' to 'Chosen so that IWC $= 10^{-6}$ kg m$^{-3}$ at all levels'. The updated table in shown in Tab. 1.1.

Table 1.1: Retrieval quantities and a priori assumptions used in the retrieval. The relation for the a priori mean of $\log_{10}(N_0^*)$ is taken from Cazenave et al. (2019).

| Quantity | Retrieved parameters | A priori mean | A priori std. dev. |
|---|---|---|---|
| Ice water content (IWC) | $\log_{10}(N_0^*)$ | $-0.076586 \cdot (T - 273.15) + 17.948$ with $T$ temperature in K | 2 |
| | $D_m$ | Chosen so that IWC $= 10^{-6}$ kg m$^{-3}$ at all levels. | 500 $\mu$m |
| Rain water content (RWC) | $\log_{10}(N_0^*)$ | 7 | 2 |
| | $D_m$ | 500 $\mu$m | 500 $\mu$m |
| Cloud liquid water content (CLWC) | $\log_{10}(\text{CLWC})$ | From ERA5 | 1 |
| Relative humidity (RH) | $\text{arctanh}(\frac{2 \cdot \text{RH}}{1.1} - 1.0)$ | From ERA5 | 1 |

**1.2 Typographical errors**

Again, we would like to thank the reviewer for pointing out these mistakes, which we will of course all correct in the revised version of the manuscript.

**Figures**

[Figure]

Figure 1.1: Like Fig. 4 from the manuscript but with modified observations errors of the radar bins affected by ground clutter set to a $10^6$.

[Figure]

Figure 1.2: Like Fig. 11 from the manuscript but with modified observations errors of the radar bins affected by ground clutter set to a $10^6$.

**2 Comments from reviewer 2**

We would like to thank the reviewer for reading our manuscript and providing helpful feedback.

**2.1 Specific comments**

**Reviewer comment 1**

Lines 69-71. It may be helpful to add the time periods of these campaigns in the text or in the table.

**Author response:**

We will include the requested information in the revised version of the manuscript.

**Changes in manuscript**

- The sentence introducing the B984 flight will be modified to:

  **Changes starting in line 66:**

  The first considered flight, designated B984,  was performed on 14 October 2016  during the North Atlantic Waveguide and Downstream Impact Experiment (NAWDEX), which took place during September and October 2016 (Schäfler et al., 2018).

- The sentence introducing the C159 and C161 flights will be modified to:

  **Changes starting in line 71:**

  The two other flights, designated C159 and C161,  were part of the PIKNMIX-F campaign, which took place in March 2019.

**Reviewer comment 2**

Line 100. It seems that you do not have the same measurements to perform the retrievals for different flights. How is the missing information handled in the retrieval method?

**Author response:**

To handle the varying availability of channels (and sensors) across the different flights, the retrieval implementation is adaptive in the sense that it can be run with arbitrary sensor configurations. Channels that are sensitive to surface emission and thus only used over Ocean are disabled by setting their assumed uncertainties to very high values.

**Changes in manuscript:**

- To make this clearer, we will reformulate the sentence that describes the adaption of the forward model and retrieval.

  > **Changes starting in line 134:**
  >
  > The  forward model and retrieval were made adaptive so that the ingested observations can be easily adapted to the  different sensors and channels that were available for each flight. Low frequency channels that are used only over Ocean surfaces are deactivated over land by setting the corresponding channel uncertainty to $10^6$ K.

**Reviewer comment 3**

Figure 10. The discrepancies are quite large for 243 GHz channel for flight C161 compared to the other two flights. Could you comment on that?

**Author response:**

As we explain in the manuscript, we suspect that the remaining discrepancies in the 243 GHz channel for flights C159 and C161 are mainly caused by precipitation that may be observed differently by the different sensors due to co-location issues. This reasoning is based on the observation that similar residuals are observed in the same location in other passive channels that are sensitive to the lower atmosphere as well as the radar. For the region where the largest biases are observed for flight C161 the temporal delay between radiometer and radar observations is about 30 minutes during which the structure of the cloud has likely changed thus leading to inconsistencies between the radar and radiometer observations.

The initial version of the manuscript has not discussed the temporal co-location of the observations that differs significantly between the flights. For the revised manuscript we propose to add a new figure that displays the time delay between the radar and radiometer observations for the different flights. We will also extend the discussion of the residuals.

**Changes in manuscript:**

- We will add the figure shown in Fig. 3.1 together with the description shown below to the the section the presents the radar observations.

**Changes starting in line 90:**

While the radar observations for flight B984 come from an airborne radar, the observations for flights C159 and C161 stem from a spaceborne sensor. The high velocity of the spaceborne sensor causes significant temporal delay between co-located observations from the radiometers and the radar. Figure 3.1 displays the delay between co-located radar and radiometer observations with respect to the along-track distance for the three flight scenes. So while the delays for flight B984 remain mostly within 5 minutes, they reach values exceeding 30 minutes for the two other flights.

- The presentation of the retrieval residuals will be extend as follows.

**Changes starting in line 188:**

Radiometer residuals for flight B984 are mostly within ±5 K  but larger for flights C159 and C161. For these two flights, residuals exceeding 10 K are observed  in the window channels up to 243 GHz as well as  the outermost channels around the absorption lines at 118 GHz and 183 GHz. Since these  occur in profiles where precipitation is present and in which similar residuals can be observed in the radar observations, a likely explanation is that they are caused by  precipitation that is not observed by all sensors due to spatial and temporal co-location issues. Especially the large residuals in the 243 GHz channel for flight C161 at around 100 km along track distance may well be caused by the evolution of the convective cloud during the delay of almost 30 minutes that separates the radiometer and radar observations.

**Reviewer comment 4**

Lines 220-222. As you also mentioned, the largest uncertainties correspond to the higher-level clouds where we have smaller ice particles. Are these uncertainties also related to the lack of representativeness of the particle shape/type used in the models?

**Author response:**

As we meant to express in the paragraph starting in l. 285, our hypothesis is that the underestimation of IWC high up in the cloud that is observed across all tested particle models is rather due to the mismatch in shape between the assumed and observed PSD. Since none of the sensors used in this study has any significant sensitivity to particles with diameters smaller than 200 $\mu$m, the contribution of those particles to the total

[Figure]

Figure 2.1: Delays between the co-located observations from radar and radiometers for the three flights.

IWC is essentially inferred through the assumed shape of the PSD. Since the assumed PSD drastically underestimates the amount of those particles, this may explain why the retrieved IWC is lower than the true IWC.

**Changes in manuscript:**

To make this point clearer, we will reformulate the paragraph that discusses the deviations from the in situ measurements.

> **Changes starting in line 288:**
>
> This indicates that the  concentrations of these larger particles may be retrieved correctly but that the total IWC is underestimated due to the mismatch between assumed and actual PSD shape, the former of which lacks the very high concentration of small particles that are present in the in situ measurements. Although O'Shea et al. (2021) and O'Shea et al. (2019) show that the  occurrence of high particle concentrations at sizes below 200 $\mu$m may be due to measurement inaccuracies of the CIP-15 probe, the measured PSDs correctly reproduce the measured IWC at these altitudes when the corresponding water content is calculated using any of the tested particle habits (Fig. 8).

**3 Comments from reviewer 3**

We would like to thank the referee for reading our manuscript and providing helpful feedback.

**3.1 Specific comments**

**Reviewer comment 1**

Line 69: Please include the time periods of these campaigns for completeness.

**Author response:**

We will include the requested information in the revised version of the manuscript.

**Changes in manuscript:**

- The sentence introducing the B984 flight will be modified to:

  > **Changes starting in line 66:**
  >
  > The first considered flight, designated B984,  was performed on 14 October 2016 as part of the North Atlantic Waveguide and Downstream Impact Experiment (NAWDEX), which took place during September and October 2016 (Schäfler et al., 2018).

- The sentence introducing the C159 and C161 flights will be modified to:

  > **Changes starting in line 71:**
  >
  > The two other flights, designated C159 and C161,  were part of the PIKNMIX-F campaign, which took place in March 2019.

**Reviewer comment 2**

Line 107 and 108: Title 2.2 missing capital letter (In situ measurements). Also missing the capital letter in the first sentence of the paragraph below.

**Author response:**

We will correct this in the revised version of the manuscript.

**Reviewer comment 3**

Line 109: For completeness I would recommend a better description of what it is meant by "high-level" and the "lower parts" of in situ-sampling. Perhaps it makes more sense to introduce in-situ measurements before the analysis of the co-location of flight tracks.

**Author response:**

We agree with the reviewer that introducing the in-situ measurements before the analysis of the co-location of flight tracks does indeed improve the structure of the manuscript.

**Changes in manuscript:**

We will shorten and merge the paragraph starting in l. 112 with the paragraph starting in l. 108.

> **Changes starting in line 108:**
>
> The in situ measurements  that are relevant to this study are measurements of bulk ice water content using a Nevzorov hot-wire probe (Korolev et al., 2013) and PSDs recorded using DMT CIP-15 and CIP-100 probes, which measure size-resolved particle concentrations with resolutions of 15 and 100 $\mu$m, respectively. In situ measurements are available only for flights B984 and C159, which each consist of two parts: A high level run during which the aircraft flew above the cloud system to perform the remote sensing observations and a low level run during which the aircraft flew at lower altitude through the cloud to perform the in situ measurements. A detailed view of the  high and low level runs for the two flights are provided in Fig. 5. For flight C159, this view reveals a noticeable horizontal offset of 3 to 4 km between the ground  tracks of radar and radiometer observations . Even larger deviations occur between certain parts of the  low level run and the ground tracks of the remote sensing observations.

**Reviewer comment 4**

Line 130: Background properties of the atmosphere and the surface [. . . ]

**Author response:**

We will correct this for the revised version of the manuscript.

**Reviewer comment 5**

Line 132-133: Although readers are referred to Pfreundschuh et al. (2020) for a detailed description of the retrieval, perhaps a better synergy between the text and table 2 would add to the description here regarding the parameters of the PSD for the different species.

**Author response:**

To improve the description of the retrieved PSD parameters as well as the following description of the retrieval, we will extend the paragraph describing the retrieval outputs to introduce the mathematical form of the PSD. The revised version of the paragraph reads:

**Changes in manuscript:**

> **Changes starting in line 131:**
>
> The output of the retrieval are two parameters of the PSDs of frozen and liquid hydrometeors as well as liquid cloud water content (LCWC) and relative humidity.  Hydrometeor PSDs are represented using the approach proposed by Delanoë et al. (2005): At each level in the atmosphere the concentration of hydrometeors with respect to the volume equivalent diameter $D_{\mathrm{eq}}$ is given by
>
> $$N(D_{\mathrm{eq}}) = N_0^* F(\frac{D_{\mathrm{eq}}}{D_m}) \qquad (1)$$
>
> where $F$ is a fixed function that specifies the shape of the normalized PSD and $N_0^*$ and $D_m$ are the retrieved parameters. The $N_0^*$ parameter is retrieved in log space while $D_m$ is retrieved in linear space. Relative humidity is retrieved in a transformed space based on an inverse hyperbolic tangens transformation and CLWC in log space. A listing of all retrieval targets and corresponding a priori assumptions  is provided in Tab. 2.

**Reviewer comment 6**

Line 136: [...] Atmospheric Radiative Transfer Simulator (ARTS, Buehler et al., 2018) is used [...]

**Author response:**

Since the ARTS acronym is introduced already earlier in the manuscript we will rewrite the sentence as follows.

**Changes in manuscript:**

> **Changes starting in line 135:**
>
> The latest stable release (version 2.4) of  ARTS (Buehler et al., 2018) is used to implement the forward model used in the retrieval.

**Reviewer comment 7**

Line 145: "In the first one, the bulk properties". Use lower case in "in the first one [...]"

**Author response:**

We will correct this in the revised version of the manuscript.

**Reviewer comment 8**

Line 151: After the sentence that starts with "The updated values of [...]", for completeness perhaps the cloud ice PSD equation could be included in parentheses with the Dm, N0* and the shape parameter.

**Author response:**

The ice PSD equation in its complete form looks as follows:

$$N(D_{\text{eq}}) = N_0^* \beta \frac{\Gamma(4)}{4^4} \frac{\Gamma(\frac{\alpha+5}{\beta})^{4+\alpha}}{\Gamma(\frac{\alpha+4}{\beta})^{5+\alpha}} (\frac{D_{\text{eq}}}{D_m})^\alpha \exp\left\{ -\left( \frac{D_{\text{eq}}}{D_m} \frac{\Gamma(\frac{\alpha+5}{\beta})}{\Gamma(\frac{\alpha+4}{\beta})} \right)^\beta \right\} \qquad (3.1)$$

Due to its relatively bulky form and since we felt it does not contribute any useful information for the interpretation of the presented results, we chose not to reproduce the equation in the manuscript
However, in order to make the role of the shape parameters mentioned in the manuscript more clear we will rewrite the sentence in question and refer to the compact form of the PSD that will be included in the revised version of the manuscript.

**Changes in manuscript:**

The  normalized shape function $F$ in Eq. (1) follows a modified gamma distribution shape using the parameters from Cazenave et al. (2019).

**Reviewer comment 9**

Line 152: single particle optical properties.

**Author response**

We will correct this in the revised version of the manuscript.

**Reviewer comment 10**

Line 154: "Since this is difficult". Please expand on this for completeness.

**Author response**

We will expand on the difficulty of choosing a particle model for ice hydrometeors in the revised manuscript.

**Changes in manuscript:**

> **Changes starting in line 154:**
>
>  Due to the large variability of ice particle shapes in real clouds, it is unclear which particle habit should be chosen to best represent their radiative properties or whether such a unique best model exists at all. Hence, the approach taken here is to select a set of habits  and perform the retrieval with each of them. This will allow us to investigate the impact of the selected habit on the retrieval results.

**Reviewer comment 11**

Line 175: retrieved hydrometeor size distributions.

**Author response:**

We will correct this in the revised version of the manuscript.

**Reviewer comment 12**

Line 224: for which the best agreement.

**Author response:**

We will correct this in the revised version of the manuscript.

**Reviewer comment 13**

Line 286: typo in "For, flight B984 [...]". Move the comma please.

**Author response:**

We will correct this in the revised version of the manuscript.

**Reviewer comment 14**

Line 298: typo in "Secondly, the a clear backscattering"

**Author response:**

We will incorporate this suggestion in the revised version of the manuscript.

**Reviewer comment 15**

Line 349: Just comment here. I wonder if for a real scenario, with the complexities of the vertical differences in particle orientation, habit phase, etc, this would add to 20% in the resultant observations at nadir.

**Author response:**

This is certainly true. However, the point we were trying to make was to estimate the impact that neglecting particle orientation may have on the presented results. We were therefore interested only in an upper bound of the effect that particle orientation may have on the results in order to estimate the robustness of our results.

**Reviewer comment 16:**

Figure 10. The discussion mentions the large differentes for 243 GHz, however flight B984 shows smaller residuals at this channel than the other two channels. Is there anything to add to the discussion regarding this? Also, there are large differentes for 448 +- 7.2 GHz, specially for flight C159. Could you comment on that too?

**Author response:**

As we explain in the manuscript, we suspect that the larger residuals for flights C159 and C161 are caused by precipitation that is not observed by both sensors due to co-location issues. In addition to spatial co-location issues, flight C159 and C161 are affected by temporal co-location issues with delays of up to 30 minutes between the observations.
The initial version of the manuscript has not discussed the temporal co-location of the observations that differs significantly between the flights. For the revised manuscript we propose to add a new figure that displays the time delay between the radar and radiometer observations for the different flights. We will also extend the discussion of the residuals.
While the 448 $\pm$ 7.2 GHz channel may also be affected by this, the residuals look mostly random and are thus most likely caused by the increased thermal noise in this channel.

**Changes in manuscript:**

We will rewrite the discussion of the retrieval residuals.

- We will add the figure shown in Fig. 3.1 together with the description shown below to the the section the presents the radar observations.

  **Changes starting in line 90:**

  While the radar observations for flight B984 come from an airborne radar, the observations for flights C159 and C161 stem from a spaceborne sensor. The high velocity of the spaceborne sensor causes significant temporal delay

between co-located observations from the radiometers and the radar. Figure 3.1 displays the delay between co-located radar and radiometer observations with respect to the along-track distance for the three flight scenes. While the delays for flight B984 remain mostly within 5 minutes, they reach values exceeding 30 minutes for the two other flights.

- The presentation of the retrieval residuals will be extend as follows.

    **Changes starting in line 188:**

    Radiometer residuals for flight B984 are mostly within ±5 K    but larger for flights C159 and C161. For these two flights, residuals exceeding 10 K are observed  in the window channels up to 243 GHz as well as  the outermost channels around the absorption lines at 118 GHz and 183 GHz. Since these  occur in profiles where precipitation is present and in which similar residuals can be observed in the radar observations, a likely explanation is that they are caused by  precipitation that is not observed by all sensors due to spatial and temporal co-location issues. Especially the large residuals in the 243 GHz channel for flight C161 at around 100 km along track distance may well be caused by the evolution of the convective cloud during the delay of almost 30 minutes that separates the radiometer and radar observations.

[Figure]

Figure 3.1: Delays between the co-located observations from radar and radiometers for the three flights.

**Reviewer comment 17:**

Figure A1: Have you looked at a similar figure for the other channels?

**Author response:**

We did indeed look at similar figures for other channels but focused on channels that were present on all flights, of which none showed any indications of a relationship between IWP and residuals. However, when revisiting these plots, we discovered an error in Fig. A1 from the manuscript, which showed residuals from the $325 \pm 1.5$ GHz channel instead of the $325 \pm 3.5$ GHz channel for flight B984. We will of course correct this for the revised version of the manuscript.

Moreover, upon closer inspection of the residuals in the different channels that were available for flight B984, we did discover signs of a potential effect of the assumed ice particle shape on the retrieval residuals. We therefore propose to include an additional figure with scatter plots for these channels in the manuscript.

**Changes in manuscript:**

- We will replace Fig. A1 with the corrected version shown in Fig. 3.2.

- We will include the figure shown in Fig. 3.3 in the manuscript.

- We will extend the discussion of the impact of the ice particle shape on the residuals as follows.

  **Changes starting in line 264:**

  In an effort to better separate a potential signal from the ice particle shape in the retrieval residuals, we have investigated the relationship between retrieved IWP and the residual for different channels. Most channels that were available on all flights do not show a clear sign of a relation between the particle shape and the residuals. As an example for those channels we provide scatter plots of the retrieved IWP and the channel residual for the $325 \pm 3.5$GHz channel in Fig. 3.2 in the appendix. We did however identify two channels from flight B984 that may exhibit a potential signal from the ice particle shape in the residuals. The scatter plots for these two channels are provided in Fig. 3.3. For the $325 \pm 9.5$ GHz channel, all tested particles except the Large Plate Aggregate seem to manifest a positive correlation between IWP and  the residuals. For the $243 \pm 2.5$GHz, the 6-Bullet Rosette, 8-Column Aggregate and Large Plate Aggregate exhibit a weak negative trend in the residuals, while it remains positive for the Large Column Aggregate and Evans Snow Aggregate. At least for these two channels the Large Plate Aggregate seems to stand out as the ice particle shape yielding the smallest residuals across the retrieved range of IWP values.

  We will adapt all other sections that discuss the ability of the sub-millimeter

observations to constrain the shape of ice particles to take the this potential signal into account.

Since the Large Plate Aggregate is the particle for which the best agreement between retrieved and in situ measurements was obtained, this may be viewed as an encouraging result indicating that sub-millimeter observations can, at least in combination with radar observations, be used to constrain the shape of ice particles in clouds. However, taking into account that these are observations from only one flight as well as the complicated statistics of the results from Fig. 3.3, it remains unclear whether these findings are statistically significant. A potential confounding factor may be the impact of the a priori assumptions on these results. Since the retrieval balances the residual with the deviation from the a priori, this may lead to a worse fit for the softer particles (Large Column Aggregate, Evans Snow Aggregate) for which a much higher $D_m$ must be retrieved for a similar scattering effect. While this effect may be desired in the retrieval to avoid the apparently excessive amounts of ice retrieved using these particle shapes, it is the combination of observations and a priori assumptions that constrains the particle shape and not the observations alone. We present these results here mainly for completeness and to serve as a potential basis for further investigation.

**Reviewer comment 18:**

Figure 14: How do these results translate to IWP? Perhaps a general summary of such a Figure could add to the discussion of Figure 14.

**Author response:**

The IWP along the in situ measurement path is mostly dominated by the high concentrations at the base of the cloud. This leads to a consistent overestimation of the IWP for the radar only retrieval. The combined retrieval exhibits large variability in the results but the Large Plate Aggregate and 6-Bullet Rosette yield the results closest to the in situ measurements.

**Changes in manuscript:**

- We will include Tab. 3.1 in the revised manuscript, which contains the in situ measured and retrieved IWP for flight B984.

- We will extend the discussion of the added value of the combined retrieval as follows.

  **Changes starting in line 327:**

  The tendencies observed for the retrieved IWC in Fig. 14 are even more pro-

[Figure]

Figure 3.2: Scatter plots of retrieved IWP and corresponding residual in the fitted observations for the $325 \pm 3.5\,\mathrm{GHz}$ ISMAR channel. Each column displays the residual distributions for the five different particle habits. The gray line in each panel represents the regression line for the plotted data points. The text displays the correlation coefficient $r$ and the $p$ value of a two sided significance test for the slope of regression line.

[Figure]

Figure 3.3: Brightness temperature residuals between true and simulated observations for two channels from flight B984. The first row shows the results for $243 \pm 2.5$ GHz channel, while the second row shows the results for the $325 \pm 9.5$ GHz channel. Columns show the results for the 5 tested particles shapes. The gray line in each panel represents the regression line for the plotted data points. The text displays the correlation coefficient $r$ and the $p$ value of a two sided significance test for the slope of regression line.

nounced when the IWP is calculated along the sampling path of the in situ measurements. The resulting retrieved IWP values are displayed in Tab. 3.1. The radar only retrieval systematically overestimates the reference IWP for all tested particle shapes. The combined retrieval leads to even stronger overestimation when the Large Column Aggregate or the Evans Snow Aggregate are used as ice particle shapes, while the 8-Column Aggregate leads to a strong underestimation of the true IWP. With the 6-Bullet Rosette and the Large Plate Aggregate as ice particle shapes the combined retrieval yields results that are closest to the in situ measurements. Thus, while the incorporation of passive observations increases the sensitivity to the representation of hydrometeors, it can help to improve the retrieval of IWP given that a suitable particle model is used in the retrieval.

**Reviewer comment 17**

Figure 15: I have a tough time with the gray of in-situ (sample)

**Author response**

The legend for Fig. 15 erroneously included an entry for gray lines that were labeled 'in situ (sample)'.

Table 3.1: Retrieved IWP along in situ flight path for flight B984 for the combined and radar-only retrieval.

| | IWP [kg m$^{-3}$] | |
|---|---|---|
| **Habit** | Combined | Radar-only |
| 6-Bullet Rosette | 0.3362 | 0.4971 |
| 8-Column Aggregate | 0.2383 | 0.6783 |
| Large Column Aggregate | 0.7868 | 0.4116 |
| Large Plate Aggregate | 0.3666 | 0.4664 |
| Evans Snow Aggregate | 0.7082 | 0.5073 |
| **In situ** | | 0.3615 |

**Changes in manuscript:**

We will remove the entry from the figures legend. The corrected plot is shown in Fig. 3.4.

**Reviewer comment 18**

Figure 14 and Figure 15. Results are presented for flight B984, which uses the HAMP MIRA 35 GHz cloud radar. Could you comment on possible differences with CloudSat CPR were to be used.

**Author response:**

Although we did not investigate the effect of the radar frequency on the synergy between radiometer and radar observations in detail, we do not expect the results to change significantly if instead of the HAMP MIRA radar the CloudSat CPR were to be used. Although the higher frequency of the CloudSat radar should yield a relatively higher sensitivity to small particles, which may in crease the sensitivity to the assumed ice habit, the HAMP MIRA radar has the advantage of having higher absolute sensitivity. Furthermore, due to its lower frequency and airborne deployment, multiple scattering effects can be neglected for the MIRA radar, which is not the case for CloudSat CPR in a spaceborne configuration.
Regardless of the specific radar used, the main difficulty of determining ice concentrations in the cloud is that a radar only retrieval has only one piece of information per radar bin to infer the two moments of the hydrometeor distribution and thus has to rely on a priori information which cannot always accurately represent the properties of the observed hydrometeors.

**Changes in manuscript:**

We will add the following paragraph to the discussion of the added value of the combined retrieval.

[Figure]

Figure 3.4: In situ measured and retrieved PSDs for flight B984 retrieved using the combined (panel (a)) and the radar-only retrieval (panel (b)). Each row of panels shows the mean of the in situ measured PSDs (black) together with randomly drawn samples of measured PSDs (light grey) for a given altitude bin of a height of one kilometer. Colored lines on top show the corresponding mean retrieved PSD for different assumed particle shapes.

**Changes starting in line 319:**

While these results were obtained for a Ka band cloud radar, we do not expect them to change much for a W band radar especially if it is spaceborne. Although the habit may have a stronger effect on the retrieval results of a W band radar due to its higher frequency, the underlying problem remains that the radar observations provide only a single piece of information per range bin. To retrieve the two moments of the hydrometeor PSD, the retrieval thus has to rely on a priori information which cannot accurately describe the distributions in all clouds.

**4 Marked-up differences**

[revised manuscript text omitted]
 a spaceborne sensor. The high velocity of the spaceborne sensor causes significant temporal delay between co-located observations from the radiometers and the radar. Figure 3 displays the delay between co-located radar and radiometer observations with respect to the along-track distance for the three flight scenes. While the delays for flight B984 remain mostly within 5 minutes, they reach values exceeding 30 minutes for the two other flights.

**2.1.1 MARSS**

The MARSS radiometer measures microwave radiances at 89 GHz, 157 GHz and channels located around the water vapor line at 183 GHz. Although MARSS is a scanning radiometer only observations within 5 ° off nadir are used in the retrieval. The

**Table 1.** Datasets used in the this study.

| Title | Usage | Reference |
|---|---|---|
| HALO Microwave Package measurements during North Atlantic Waveguide and Downstream impact EXPeriment (NAWDEX) | Radar observations for for flight B984 | Konow et al. (2018) |
| CloudSat 1B-CPR | Radar observations for flight C159 (Granule 67658), C161 (Granule 68702) | Tanelli et al. (2008) |
| FAAM B984 ISMAR and T-NAWDEX flight: Airborne atmospheric measurements from core instrument suite on board the BAE-146 aircraft | Radiometer observations and in situ measurements for flight B984 | Facility for Airborne Atmospheric Measurements (2016) |
| FAAM C159 PIKNMIX-F flight: Airborne atmospheric measurements from core and non-core instrument suites on board the BAE-146 aircraft | Radiometer observations and in situ measurements for flight C159 | Facility for Airborne Atmospheric Measurements (2019a) |
| FAAM C161 PIKNMIX-F flight: Airborne atmospheric measurements from core and non-core instrument suites on board the BAE-146 aircraft | Radiometer observations for flight C159 | Facility for Airborne Atmospheric Measurements (2019b) |
| ERA5 global reanalysis | A priori state and atmospheric background fields | Hersbach et al. (2018) |

100 observations from the three flights are displayed in Fig. 4. Observations from channels that are sensitive to surface emission (89 GHz and 157 GHz) are excluded from the retrieval for flight sections over land. The MARSS observations were mapped to the radar observations using nearest-neighbor interpolation.

**2.1.2 ISMAR**

The ISMAR radiometer has channels covering the frequency range from 118 GHz up to 874 GHz. As for MARSS, only
105 observations within $5°$ degrees off nadir are used in the retrieval. The observations from the 3 flights are displayed in Fig. 5. Similar as for the two low-frequency channels of MARSS, the 4 outermost channels around the 118 GHz oxygen line are not used over land. The matching of ISMAR observations to radar observations is performed in the same way as for MARSS. It should be noted that not all channels were available on all flights: The channels around 448 GHz were not available on the B984 flight, while the two of the channels around 325 GHz were missing for the C159 and C161 flights. From the channels at
110 874 GHz only the V polarization was available for flights C159 and C161.

[Figure]

**Figure 2.** Radar observations from the flights used in this study. Panel (a) shows the radar reflectivity measured by the HAMP MIRA 35 GHz cloud radar. Panels (b) and (c) show the reflectivity measured by the CloudSat CPR at 94 GHz. The white line displays the ERA5 freezing level from Hersbach et al. (2018).

The polarized measurements at 243 GHz and at 664 GHz for flight B984 were replaced by the average of the measured H and V polarizations. For flights C159 and C161, only the horizontally-polarized measurements at 664 GHz were used used due to excessive noise in the V channel.

**2.2   In situ measurements**

115   The in situ measurements  that are relevant to this study are measurements of bulk ice water content using a Nevzorov hot-wire probe (Korolev et al., 2013) and PSDs recorded using DMT CIP-15 and CIP-100 probes, which measure size-resolved particle concentrations with resolutions of 15 and 100 μm, respectively. In situ measurements are available only for flights B984 and C159, which each consist of two parts: A high level run during which the aircraft flew above the cloud system to perform the remote sensing observations and a low level run during which the aircraft

120   flew at lower altitude through the cloud to perform the in situ measurements. A detailed view of the

[Figure]

**Figure 3.** Delays between the co-located observations from radar and radiometers for the three flights.

[Figure]

**Figure 4.** Passive microwave measurements from the MARSS radiometer together with the matched radar observations. Grey background in the radiance plots marks observations that were taken over land.

 high and low level runs for the two flights are provided in Fig. 6. For flight C159, this view reveals a noticeable horizontal offset of 3 to 4 km between the ground  tracks of radar and radiometer observations

[Figure]

**Figure 5.** Passive microwave measurements from the ISMAR radiometer together with the matched radar observations. Grey background in the radiance plots marks observations that were taken over land and are therefore not used in the retrieval.

 . Even larger deviations occur between certain parts of the low level run and the ground tracks of the remote sensing observations.

125 ~~The in situ measurements that are relevant to this study are bulk ice water content measured using a Nevzorov hot-wire probe (Korolev et al., 2013) and PSDs recorded using DMT CIP-15 and CIP-100 probes, which measure size-resolved particle concentrations with resolutions of 15 and 100 μm, respectively. The in situ measurements were mapped to corresponding radar observations using a nearest-neighbor criterion.~~

[Figure]

**Figure 6.** Detailed view of the flight paths of the high-level runs and in situ sampling paths for flights B984 and C159. The backound is the true-color composite derived from the closest overpasses of the MODIS (Team, 2017) sensor on the Aqua satellite.

An overview of the measured IWC and PSDs is provided in Fig. 7. While for flight B984 the measured IWC are mostly
130 consistent with the radar observations, there are clear disparities between the measured IWC and the CPR reflectivities for flight C159. This indicates that there may be considerable differences between the regions of the cloud that were sampled during the in situ sampling and the part the was observed by the CloudSat CPR.

The PSD profiles for flight B984 show a clear size-sorting pattern with a gradual decrease of the concentration of particles smaller than 200 µm and a simultaneous increase of the concentration of larger particles. For flight C159, high concentrations
135 of small particles are encountered at low altitudes which decrease with altitude. For larger particles no systematic variation with altitude is observed.

**2.3 Retrieval algorithm**

The synergistic retrieval algorithm used in this study is based on the optimal estimation framework (Rodgers, 2000) and retrieves distributions of frozen and liquid hydrometeors together with water vapor by simultaneously fitting a forward model
140 to the active and passive observations. Since the algorithm is described in detail in Pfreundschuh et al. (2020) the following section only outlines its main features and how it has been adapted to the flight data.

The retrieval input consists of a single radar profile and the corresponding spatially closest radiometer observations. Background properties of  the atmosphere and the surface, such as temperature and wind speed, as well as a priori profiles for relative humidity and liquid cloud water are taken from the ERA5 hourly reanalysis (Hersbach et al., 2018). The
145 output of the retrieval are two parameters of the PSDs of frozen and liquid hydrometeors as well as liquid cloud water content (LCWC) and relative humidity.  Hydrometeor PSDs are represented using the approach proposed by Delanoë et al. (2005): At each level in the atmosphere the concentration of hydrometeors with respect to the volume equivalent diameter $D_{\mathrm{eq}}$ is given

[Figure]

**Figure 7.**  In situ measured IWC and PSDs for flights B984 and C159. The first row of panels displays the measured IWC along the flight path plotted on top of the co-located radar observations. The second row displays the variation of the mean of the in situ measured PSDs for different altitudes in the cloud.

by

$$N(D_{\text{eq}}) = N_0^* F\left(\frac{D_{\text{eq}}}{D_m}\right) \tag{1}$$

150   where $F$ is a fixed function that specifies the shape of the normalized PSD and $N_0^*$ and $D_m$ are the retrieved parameters. The $N_0^*$ parameter is retrieved in log space while $D_m$ is retrieved in linear space. Relative humidity is retrieved in a transformed space based on an inverse hyperbolic tangens transformation and CLWC in log space. A listing of all retrieval targets and corresponding a priori assumptions  is provided in Tab. 2.

The  forward model and retrieval were made adaptive so that the ingested observations can
155   be  adapted to the  different sensors and channels that were available for each flight. Low frequency channels that are used only over Ocean surfaces are deactivated over land by setting the corresponding channel uncertainty to $10^6$ K. The atmospheric grid was limited to altitudes between 0 and 10 km and matched to the resolution of the radar observations. The latest stable release (version 2.4) of  ARTS (Buehler et al., 2018) is used to implement the forward model used in the retrieval. The built-in single-scattering radar solver

**Table 2.** Retrieval quantities and a priori assumptions used in the retrieval. The relation for the a priori mean of $\log_{10}(N_0^*)$ is taken from Cazenave et al. (2019).

| Quantity | Retrieved parameters | A priori mean | A priori std. dev. |
|---|---|---|---|
| Ice water content (IWC) | $\log_{10}(N_0^*)$ | $-0.076586\cdot(T-273.15)+17.948$ with $T$ temperature in K | 2 |
| | $D_m$ |  Chosen so that IWC $= 10^{-6}$ kg m$^{-3}$ at all levels. | 500 µm |
| Rain water content (RWC) | $\log_{10}(N_0^*)$ | 7 | 2 |
| | $D_m$ | 500 µm | 500 µm |
| Cloud liquid water content (CLWC) | $\log_{10}(\text{CLWC})$ | From ERA5 | 1 |
| Relative humidity (RH) | $\text{arctanh}(\frac{2\cdot\text{RH}}{1.1}-1.0)$ | From ERA5 | 1 |

160 of ARTS is used to calculate radar observations and Jacobians. To account for the effect of multiple scattering in CloudSat observations, the attenuation due to hydrometeors is scaled at each atmospheric layer by a factor of $0.5$ following Fig. 16 in Battaglia et al. (2010). Passive radiances are calculated using the ARTS interface to DISORT (Stamnes et al., 2000) and their Jacobians are approximated using a first order scattering approximation. Gaseous absorption is modeled using the absorption models from Rosenkranz (1993) for $N_2$ and $O_2$. Following Fox (2020), absorption from water vapor is calculated using

165 a combination of the AER database v3.6 (Cady-Pereira et al., 2020) for resonant absorption and the MT-CKD model version 3.2 for continuum absorption (Mlawer et al., 2012).

**2.3.1 Representation of frozen hydrometeors**

The forward model simulates active and passive observations in two steps: In the first one, the bulk properties that are used to represent hydrometeors in the retrieval are mapped to corresponding optical properties. The optical properties are then, in the

170 second step, used together with background atmosphere and surface to simulate the observations.

The mapping of bulk to optical properties is based on a PSD and an ice particle habit that associates particles of different sizes and shapes to optical properties. As described above, the forward model uses the normalized PSD approach proposed by Delanoë et al. (2005) with the mass-weighted mean diameter ($D_m$) and intercept parameter ($N_0^*$) as parameters. The  normalized shape function

175 $F$ in Eq. (1) follows a modified gamma distribution shape using the parameters from Cazenave et al. (2019). The ice particle habit is represented by a collection of ice particle shapes and corresponding, pre-computed single particle optical properties. Bulk optical properties are calculated by integrating the product of particle density and optical properties over the particle size. As the retrieval is currently set up, the particle habit cannot be retrieved and must be assumed a priori.

 Due to the large variability of ice particle shapes in real clouds, it is unclear which particle habit should be chosen to best represent their radiative properties or whether such a unique best model exists at all. Hence, the approach taken here is to select a set of habits  and perform the retrieval with each of them. This will allow us to investigate the impact of the selected habit on the retrieval results.

[revised manuscript text omitted]

Radiometer residuals for flight B984 are mostly within ±5 K  but larger for flights C159 and C161. For these two flights, residuals exceeding 10 K are observed  in the window channels up to 243 GHz as well as  the outermost channels around the absorption lines at 118 GHz and 183 GHz. Since these  occur in profiles where precipitation is present and  in the radar observations and other channels that are sensitive to the lower parts of the atmosphere, a likely explanation is that they are caused by  precipitation that is not observed by all sensors due to spatial and temporal co-location issues. Especially the large residuals

in the 243 GHz channel for flight C161 at around 100 km along track distance may well be caused by the evolution of the convective cloud during the delay of about 30 minutes that separates the radiometer and radar observations.

[revised manuscript text omitted]
, we have investigated the relationship between retrieved IWP and the residual for different channels. Most channels that were available on all flights do not show a clear sign of a relation between the particle shape and the residuals. As an example for those channels we provide scatter plots of the retrieved IWP and the channel residual for the $325 \pm 3.5\,$GHz channel in Fig. A1 in the appendix . We did however identify two channels from flight B984 that may exhibit a potential signal from the ice particle shape in the residuals. The scatter plots for these two channels are provided in Fig. 15. For the $325 \pm 9.5$ GHz channel, all tested particles except the Large Plate Aggregate seem to manifest a positive correlation between IWP and  the residuals. For the $243 \pm 2.5\,$GHz, the 6-Bullet Rosette, 8-Column Aggregate and Large Plate Aggregate exhibit a weak negative trend in the residuals, while it remains positive for the Large Column Aggregate and Evans Snow Aggregate. At least for these two channels  Large Plate Aggregate seems to stand out as the ice particle shape yielding the smallest residuals across the retrieved range of IWP values.

Since the Large Plate Aggregate is one of the particles for which the best agreement between retrieved and in situ measurements was obtained, this may be viewed as an encouraging result indicating that sub-millimeter observations can, at least in combination with radar observations, be used to constrain the shape of ice particles in clouds. However, taking into account that these are observations from only one flight as well as the complicated statistics of the results from Fig.  15, it remains unclear whether these findings are statistically significant. A potential confounding factor may be the impact of the a priori assumptions on these results. Since the retrieval balances the residual with the deviation from the a priori, this may lead to a worse fit for the softer particles (Large Column Aggregate, Evans Snow Flake) for which a much higher $D_m$ must be retrieved for a similar scattering effect. While this effect may be desired in the retrieval to avoid the apparently excessive amounts of ice retrieved using these particle shapes, it is the combination of observations and a priori assumptions that constrains the particle shape and not the observations alone. We present these results here mainly for completeness and to serve as a potential basis for further investigation.

Nonetheless, even if indeed present, a potential signal from the ice particle shape in the results would be limited to a few Kelvin. This implies that future ice hydrometeor retrievals that make use of millimeter and sub-millimeter microwave observations must either account for the uncertainty caused by variations in ice particle shape or find ways to more accurately

[Figure]

**Figure 15.** Brightness temperature residuals between true and simulated observations for two channels from flight B984. The first row shows the results for the $243 \pm 2.5$ GHz channel, while the second row shows the results for the $325 \pm 9.5$ GHz channel. Columns show the results for the 5 tested particles shapes. The gray line in each panel represents the regression line for the plotted data points. The text displays the correlation coefficient $r$ and the $p$ value of a two sided significance test for the slope of regression line.

[revised manuscript text omitted]

The tendencies observed for the retrieved IWC in Fig. 14 are even more pronounced when the IWP is calculated along the sampling path of the in situ measurements. The resulting retrieved IWP values are displayed in Tab. 4. The radar only retrieval systematically overestimates the reference IWP for all tested particle shapes. The combined retrieval leads to even stronger overestimation when the Large Column Aggregate or the Evans Snow Aggregate are used as ice particle shapes, while the 8-Column Aggregate leads to a strong underestimation of the true IWP. With the 6-Bullet Rosette and the Large Plate Aggregate used as ice particle shapes, the combined retrieval yields results that are closest to the in situ measurements. Thus, while the incorporation of passive observations increases the sensitivity to the representation of hydrometeors, it can help to improve the retrieval of IWP given that a suitable particle model is used in the retrieval.

[Figure]

**Figure 17.**  In situ measured and retrieved PSDs for flight B984 retrieved using the combined (panel (a)) and the radar-only retrieval (panel (b)). Each row of panels shows the mean of the in situ measured PSDs (black) together with randomly drawn samples of measured PSDs (light grey) for a given altitude bin of a height of one kilometer. Colored lines on top show the corresponding mean retrieved PSD for different assumed particle shapes.

**Table 4.** Retrieved IWP along in situ flight path for flight B984 for the combined and radar-only retrieval.

|  | IWP [kg m$^{-2}$] | |
| --- | --- | --- |
| **In situ** | 0.3615 | |
| **Habit** | Combined | Radar-only |
| 6-Bullet Rosette | 0.3362 | 0.4971 |
| 8-Column Aggregate | 0.2383 | 0.6783 |
| Large Column Aggregate | 0.7868 | 0.4116 |
| Large Plate Aggregate | 0.3666 | 0.4664 |
| Evans Snow Aggregate | 0.7082 | 0.5073 |

These results thus suggest that combining radar with passive microwave observations helps to constrain the PSD of ice hydrometeors for sufficiently large particle sizes ($D_{\mathrm{MAX}} > 200\,\mu\mathrm{m}$). Since for air- and space-borne observations only microwave observations can sense the base of thick clouds, this is a unique synergy between these types of observations.

While these results were obtained for a Ka band cloud radar, we do not expect them to change much for a W band radar. Although the habit may have a stronger effect on the retrieval results of a W band radar due to its higher frequency, the underlying problem remains that the radar observations provide only a single piece of information per range bin. To retrieve the two moments of the hydrometeor PSD, the retrieval thus has to rely on a priori information, which cannot accurately describe the distributions in all clouds. Although the passive observations provide only a comparably small amount of additional information, our results indicate that the retrieval is able to use that to better constrain the retrieved hydrometeor distributions.

**4.6 Limitations**

[revised manuscript text omitted]

435 However, the retrieval is at the same time very sensitive to the assumed ice particle habit that is used in the retrieval forward model.  Although we found some evidence of a signal that could help to constrain the ice particle shape based  on the combination of radar and  sub-millimeter observations, it remains limited to not more than 5 Kelvin. This means that more work is needed to

find out how to effectively constrain the ice particle shape with remote sensing observations or to better constrain it a priori.

Although further work will be required, this study demonstrates the feasibility and potential of synergistic retrievals of ice hydrometeors by combining active and passive observations at millimeter and sub-millimeter wavelengths. Since the combined retrieval can better constrain the PSD of ice hydrometeors, it may be a useful tool to study the representation of clouds in NWP and climate models. Additionally, as illustrated in this study, the retrieval can be used to study the representation of ice hydrometeors in radiative transfer simulations, which will be vital to many applications of observations from upcoming sub-millimeter sensors such as ICI and the Arctic Weather Satellite (ESA, 2021).

*Code availability.* All code used to produce the results in this study is available through public repositories (Simon Pfreundschuh, 2019; Pfreundschuh, 2021).

*Data availability.* A detailed listing of the datasets that were used in this study together with their sources is provided in Tab. 1.

*Author contributions.* Simon Pfreundschuh has performed the retrieval calculations and data analysis as well as written the manuscript. Patrick Eriksson, Stefan A. Buehler, Manfred Brath, David Duncan and Simon Pfreundschuh have collaborated on the study that lead to the development of the presented algorithm. Stuart Fox, Richard Cotton, Florian Ewald have provided the flight campaign data, guidance regarding their usage and contributed to the interpretation and discussion of the retrieval results.

*Competing interests.* No competing interests are present

*Acknowledgements.* The work of SP and PE on this study was financially supported by the Swedish National Space Agency (SNSA) under grants 150/14 and 166/18.

SB was supported by the Deutsche Forschungsgemeinschaft (DFG, German Research Foundation) under Germany's Excellence Strategy — EXC 2037 'Climate, Climatic Change, and Society' — Project Number: 390683824, contributing to the Center for Earth System Research and Sustainability (CEN) of Universität Hamburg.

SB's work contributes to the Cluster of Excellence "CLICCS—Climate, Climatic Change, and Society" funded by the Deutsche Forschungsgemeinschaft DFG (EXC 2037, Project Number 390683824), and to the Center for Earth System Research and Sustainability (CEN) of Universität Hamburg."

The computations for this study were performed using several freely available programming languages and software packages, most prominently the Python language (The Python Language Foundation, 2018), the IPython computing environment (Perez and Granger, 2007),

[Figure]

**Figure A1.** Scatter plots of retrieved IWP and corresponding residual in the fitted observations for the $325 \pm 3.5\,\mathrm{GHz}$ ISMAR channel. Each column displays the residual distributions for the five different particle habits. The gray line in each panel represents the regression line for the plotted data points. The text displays the correlation coefficient $r$ and the $p$ value of a two sided significance test for the slope of regression line.

[revised manuscript text omitted]